

# DARCLOS: a cloud shadow detection algorithm for TROPOMI

Victor J. H. Trees [1,2], Ping Wang [1], Piet Stammes [1], Lieuwe G. Tilstra [1], David P. Donovan [1,2], and A. Pier Siebesma [1,2]

[1]Royal Netherlands Meteorological Institute (KNMI), De Bilt, the Netherlands
[2]Delft University of Technology, Delft, the Netherlands

**Correspondence:** Victor Trees (victor.trees@knmi.nl)

**Abstract.** Cloud shadows are observed by the TROPOMI satellite instrument as a result of its high spatial resolution as compared to its predecessor instruments. These shadows contaminate TROPOMI's air quality measurements, because shadows are generally not taken into account in the models that are used for aerosol and trace gas retrievals. If the shadows are to be removed from the data, or if shadows are to be studied, an automatic detection of the shadow pixels is needed. We present the Detection AlgoRithm for CLOud Shadows (DARCLOS) for TROPOMI, which is the first cloud shadow detection algorithm for a spaceborne spectrometer. DARCLOS raises potential cloud shadow flags (PCSFs), and actual cloud shadow flags (ACSFs). The PCSFs indicate the TROPOMI ground pixels that are potentially affected by cloud shadows based on a geometric consideration with safety margins. The ACSFs are a refinement of the PCSFs using spectral reflectance information of the PCSF pixels, and identify the TROPOMI ground pixels that are confidently affected by cloud shadows. We validate DARCLOS with true color images made by the VIIRS instrument on board of Suomi NPP orbiting in close constellation with TROPOMI on board of Sentinel 5-P. We conclude that the PCSF can be used to exclude cloud shadow contamination from TROPOMI data, while the ACSF can be used to select pixels for the scientific analysis of cloud shadow effects.

## 1 Introduction

Air quality monitoring from space using satellite spectrometers started in 1978 with the launch of the first Total Ozone Mapping Spectrometer (TOMS) instrument on board the Nimbus-7 satellite. TOMS globally measured aerosol properties and concentrations of $O_3$ and $SO_2$ in the Earth's atmosphere on a daily basis, retrieved from the Earth's reflectance of sunlight using six ultraviolet (UV) wavelength bands (Heath et al., 1975). The first high-spectral resolution spectrometer was the Global Ozone Monitoring Experiment (GOME) (Burrows et al., 1999) launched in 1995, followed by the SCanning Imaging Absorption spectroMeter for Atmospheric ChartograpHY (SCIAMACHY) (Bovensmann et al., 1999), the Ozone Monitoring Instrument (OMI) (Levelt et al., 2006), the GOME-2 A/B/C instruments (Munro et al., 2016) and, most recently, the TROPOspheric Monitoring Instrument (TROPOMI) (Veefkind et al., 2012), allowing for trace gas retrieval using differential absorption features in the spectra of the Earth's reflectance (Platt and Stutz, 2008).

The spatial resolutions of TOMS, GOME, SCIAMACHY, OMI and GOME-2 have been $50 \times 50$ km$^2$, $320 \times 40$ km$^2$, $60 \times 30$ km$^2$, $24 \times 13$ km$^2$ and $80 \times 40$ km$^2$, respectively. Those resolutions are too coarse to discern kilometer-scale clouds or cloud shadows. The pixels of those spectrometers often have been partly cloudy, such that the effects of clouds, cloud shadows





and cloud-free regions are blended. Because of the inability to distinguish between those effects and the complexity of three-dimensional (3-D) radiative transfer, state-of-art algorithms for satellite spectrometers employ one-dimensional (1-D) radiative transfer models, which neglect 3-D cloud effects on cloud-free regions inside the partly cloudy pixels or on adjacent cloud-free pixels. For example, the FRESCO (Fast REtrieval Scheme for Clouds from the Oxygen A band) cloud retrieval algorithm uses

the independent pixel approximation, and does not take into account cloud shadows (Koelemeijer et al., 2001; Wang et al., 2008). However, although cloud shadows are hardly visible on the coarse resolution measurement grids of those spectrometers, they do in principle contaminate the total radiances of the large pixels.

TROPOMI on board of the ESA Sentinel-5P satellite was launched in October 2017 and is the spaceborne spectrometer with the highest spatial resolution to date: the ground pixels have dimensions of $7.2 \times 3.6$ km$^2$ in the nadir viewing direction, and

decreased to $5.6 \times 3.6$ km$^2$ on 6 August 2019. TROPOMI provides daily global maps of aerosol properties and concentrations of $O_3$, $NO_2$, $SO_2$ and HCHO from ultraviolet-visible (UV–VIS, 267–499 nm) wavelengths, of cloud properties from near-infrared (NIR, 661–786 nm) wavelengths and concentrations of CO and $CH_4$ from shortwave infrared (SWIR, 2300–2389 nm) wavelengths. Because of its high spatial resolution and data quality, TROPOMI has, for example, shown to be able to observe local $NO_2$ emission sources such as power plants (Beirle et al., 2019), gas compressor stations (van der A et al., 2020) and

cities (Lorente et al., 2019), detailed volcanic $SO_2$ plumes (Theys et al., 2019), and $CH_4$ leakage from oil/gas fields (Pandey et al., 2019; Varon et al., 2019; Schneising et al., 2020). Recently, $NO_2$ plumes from individual ships have been identified with TROPOMI in areas where the ocean sunglint enhances the signal-to-noise (Georgoulias et al., 2020).

The small pixel size of TROPOMI also causes cloud shadows to be detectable. Cloud shadow signatures can be found along cloud edges, manifested as pixels with smaller radiances than measured in their cloud-free neighborhood. Smaller measured

radiances result in lower derived reflectance values, potentially affecting TROPOMI's air quality products. Cloud shadow effects on air quality data sets can only be studied, discarded and/or corrected if the cloud shadow contaminated pixels are identified. Individual shadow pixels may be identified manually in maps of TROPOMI data through visual inspection. However, for the automatic removal or correction of shadow contaminated data, and for the statistical analysis of shadow effects on large data sets, an automatic shadow detection is needed.

For satellite spectrometer measurements, cloud shadow detection is a new topic and will become more important with the increasing spatial resolution in future satellite spectrometer missions, such as Sentinel-5 ($7.5 \times 7.3$ km$^2$) (Pérez Albiñana et al., 2017), CO2M ($< 2 \times 2$ km$^2$) (Sierk et al., 2021) and TANGO ($300 \times 300$ m$^2$) (Landgraf et al., 2020). For high spatial resolution aerial and satellite imagers, shadow detection is not new. Shadows of buildings affect the applications of aerial images, such as urban change detection and traffic monitoring (see Adeline et al., 2013, and references therein). The screening of clouds and

their shadows is an important step in the preprocessing of satellite imager data of for example Landsat and MODIS (see Zhu et al., 2018; Wang et al., 2019). Shadows degrade the quality of the images lowering the accuracy of their applications such as land cover classification and change detection (see e.g. Yan and Roy, 2020). If cloud shadows are not screened correctly, they may be confused with dark surface features such as, for example, water bodies affecting the remote sensing performance of flood detection (Li et al., 2013).





Several approaches have been followed by aerial and spaceborne imagers to detect cloud shadows. The main approaches can be categorized into geometry-based methods (Simpson and Stitt, 1998; Simpson et al., 2000; Hutchison et al., 2009) where the shadow location is computed with known or assumed parameters of the cloud shadow geometry, and spectral-based methods (see e.g. Ackerman et al., 1998) where the shadow is determined with spectral tests applied to the measured radiance. Often, a combination of those approaches is being used, first determining the potential cloud shadow locations with one of the two

approaches and subsequently refining the shadows with the other approach (see e.g. Huang et al., 2010; Zhu and Woodcock, 2012; Sun et al., 2018). The spectral tests may consist of simple darkness thresholds, however dark surface features can easily incorrectly be interpreted as shadows. Luo et al. (2008) therefore presented a method to detect cloud shadows in MODIS images exploiting the ratio between the blue and NIR (or SWIR) spectral bands, arguing that shadows may appear more blue due to the lack of direct solar illumination. Luo et al. (2008) concluded that their method is problematic over water regions

and wetlands, because the relatively dark spectra of water and shadows are difficult to distinguish. Additionally, the blueness of shadows may depend on the shadow geometry and cloud parameters such as thickness and height.

    Unsupervised machine learning (clustering) techniques have been proposed for urban shadow detection in aerial images, but the spectral variability of the shadowed materials can complicate the choice of the number of classes (see the review of Adeline et al., 2013, and references therein). Because various cloud and land surface types may be mixed within individual

pixels of satellite imagery, Bo et al. (2020) proposed a fuzzy clustering algorithm for cloud and cloud shadow detection in Landsat images, in which pixels can belong to multiple classes with associated weighting factors. Supervised machine learning techniques (neural networks and support vector machines) have been proposed for cloud shadow detection in satellite images also (see e.g. Hughes and Hayes, 2014; Ibrahim et al., 2021), but are generally computationally expensive, require large training data sets with classified shadows (which itself is the problem to be solved), and trained classifiers may not work for new scenes

with different shadow patterns (Adeline et al., 2013; Zhu et al., 2018).

    The most suitable approach for shadow detection for a satellite imager depends on the characteristics of the instrument and its host satellite. For example, the cloud and cloud shadow detection algorithm Fmask for Landsat 4-7, introduced by Zhu and Woodcock (2012), uses for its geometry-based part the thermal band (10.4 to 12.5 $\mu$m) measuring the cloud's brightness temperature. Assuming a constant lapse rate, Fmask computes the cloud top height and projection of the cloud shadow onto

the surface. For imagers that do not have a thermal band, a range of potential cloud heights can be assumed (see Zhu et al., 2015, for the application of Fmask to Landsat-8) or the approach can be limited to spectral tests only. Parmes et al. (2017) proposed a cloud and cloud shadow detection method for Suomi NPP VIIRS only based on spectral tests avoiding the usage of a thermal band, and suggested that the method could therefore also work for Sentinel-2 which does not have a thermal band. However, the accuracy of their shadow detection was low (36.1%), with a false alarm rate of 82.7%. Goodwin et al. (2013), Zhu

and Woodcock (2014), Candra et al. (2016) and Candra et al. (2019) chose to perform spectral tests based on the reflectance differences with cloud-free historical reference images, for Landsat cloud shadow detection. Such multi-temporal shadow detection approaches generally enhance the shadow detection performance (Zhu et al., 2018), but require the availability of cloud-free seasonally dependent reference images which may be challenging for satellites with long revisit periods.





In this paper we present the Detection AlgoRithm for CLOud Shadows (DARCLOS), a fast cloud shadow detection algorithm for TROPOMI and the first cloud shadow detection algorithm for a spaceborne spectrometer. DARCLOS starts with a geometry-based computation of potential shadow locations, using the cloud fraction, cloud height, viewing and illumination geometries, and surface height stored in the already available TROPOMI $NO_2$ product and cloud product FRESCO. Climatological cloud-free surface albedo reference data is available for TROPOMI and is used to perform spectral tests refining the shadows. As TROPOMI is a spectrometer, DARCLOS exploits the spectral ranges of TROPOMI by searching in each pixel for the most optimal wavelength for shadow detection independent of surface classification. The spectral tests are only based on the darkness of shadows relative to the reference data. This means that no assumptions are made about the color of cloud shadows. We validate the algorithm with true color images of Suomi NPP VIIRS which orbits in close constellation with TROPOMI.

This paper is structured as follows. In Sect. 2, we explain the method to detect cloud shadows in TROPOMI data. In Sect. 3, we show the results of the cloud shadow detection algorithm with three case studies. In Sect. 4, the validation results are presented. In Sect. 5, we discuss the limits of the algorithm and raise several points of attention for future applications. In Sect. 6, we summarize the results and state the most important conclusions of this paper.

## 2 Method

Here, we explain the method to detect cloud shadows in TROPOMI data. We first compute the potential cloud shadow flag (PCSF), explained in Sect. 2.1, and then compute the actual cloud shadow flag (ACSF), explained in Sect. 2.2. The flowchart in Fig. 1 summarizes the algorithm setup and serves as a road map for this section. The PCSFs indicate the TROPOMI ground pixels that are potentially affected by cloud shadows based on a geometric consideration with safety margins. The ACSFs are a refinement of the PCSFs using spectral reflectance information of the PCSF pixels, and indicate the TROPOMI ground pixels that are confidently affected by cloud shadows. The PCSF can be used to exclude cloud shadow contamination from the TROPOMI Level 2 data, while the ACSF can be used to select pixels for the scientific analysis of cloud shadow effects.

### 2.1 Potential cloud shadow flag (PCSF)

The PCSFs indicate the pixels that are potentially affected by cloud shadows. The PCSF is intended to be useful for filtering any cloud shadow contaminated TROPOMI data. Therefore, the number of false negative shadow detections in the PCSF should be minimized (see Sect. 4). Hence, the PCSF shadow is an overestimation of the true shadowed area.

The PCSF is computed in two steps. First, we compute the maximum potential geometric shadow extent from the cloud, with additional safety margins. Then, we flag the area between the cloud and the maximum potential shadow extent. Both steps are explained in more detail below.

#### 2.1.1 The maximum potential shadow extent

Figure 2 illustrates the cloud shadow geometry in the local reference frame at the Earth's surface. The reference frame is equivalent to the topocentric reference frame of TROPOMI (see Loots et al., 2017), except for the $xy$-plane which is now lifted





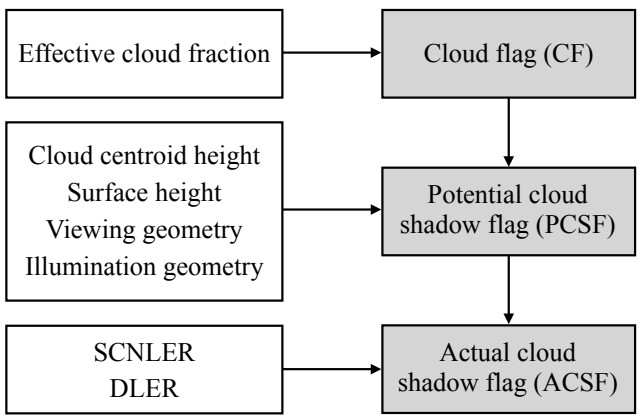

**Figure 1.** Flowchart of the algorithm. The white boxes describe the main input data and the grey boxes describe the calculated output products. SCNLER refers to the reflectivity of the scene (Sect. 2.2.1) and DLER refers to the climatological directionally dependent surface reflectivity (Sect. 2.2.2). More details are provided in the main text.

in the zenith direction with the surface height $h_{\mathrm{sfc}}$ w.r.t. the WGS84 Earth reference ellipsoid. Here, the origin (point $O$) of the reference frame is set at the center of a cloud pixel, which represents the projection of the cloud's centroid in the viewing direction onto the Earth's surface at geodetic latitude $\delta_{\mathrm{c}}$ and longitude $\vartheta_{\mathrm{c}}$. The cloud pixels are the TROPOMI ground pixels with a raised cloud flag (CF) and are determined by an effective cloud fraction (the cloud fraction assuming a cloud albedo of 0.8) larger than 0.05. The effective cloud fraction was determined in the $NO_2$ spectral window and taken from the TROPOMI

$NO_2$ data product (van Geffen et al., 2021). Angles $\theta_0$ and $\theta$ are the solar and viewing zenith angles, respectively. Angles $\varphi_0$ and $\varphi$ are the solar and viewing azimuth angles, respectively, measured positively clockwise from the North when looking in the nadir direction. The values for $\theta_0$, $\theta$, $\varphi_0$ and $\varphi$ are provided in the TROPOMI data for the origin of the *unlifted* topocentric reference frame, i.e., when $h_{\mathrm{sfc}} = 0$. In the problem of finding the cloud shadow belonging to the cloud pixel at the origin, we neglect variations of $\theta_0$, $\theta$, $\varphi_0$ and $\varphi$ in the horizontal ($x$ and $y$) and vertical ($z$) direction, and we assume that $h_{\mathrm{sfc}}$ is constant.

The location, dimensions and darkness of a cloud shadow cast on the Earth's surface and/or atmosphere below the cloud, as observed from space, may depend on (1) the cloud's location in 3-dimensional space, (2) the location of the underlying surface and/or atmosphere on which the shadow is cast in 3-dimensional space, (3) the horizontal and vertical extents of the cloud, (4) the optical thickness of the cloud, (5) the optical thickness of the atmospheric layers, (6) the illumination geometry ($\theta_0$ and $\varphi_0$), and (7) the viewing geometry ($\theta$ and $\varphi$). Because in the first step of the PCSF determination we search for the maximum

potential shadow extent, we assume an opaque cloud and neglect the optical thickness of the atmospheric layers, such that the computed shadows are cast on the Earth's surface where the shadow separation from the cloud is largest.

In Fig. 2, the cloud is located at $(x_{\mathrm{n}}, y_{\mathrm{n}}, h)$. Point $P$ at $(x_{\mathrm{n}}, y_{\mathrm{n}}, 0)$ is the nadir projection of the cloud's centroid onto the surface and $h$ is the cloud height w.r.t. the surface, which can be computed as

$$h = (1 + C)(h_{\mathrm{c}} - h_{\mathrm{sfc}}). \tag{1}$$



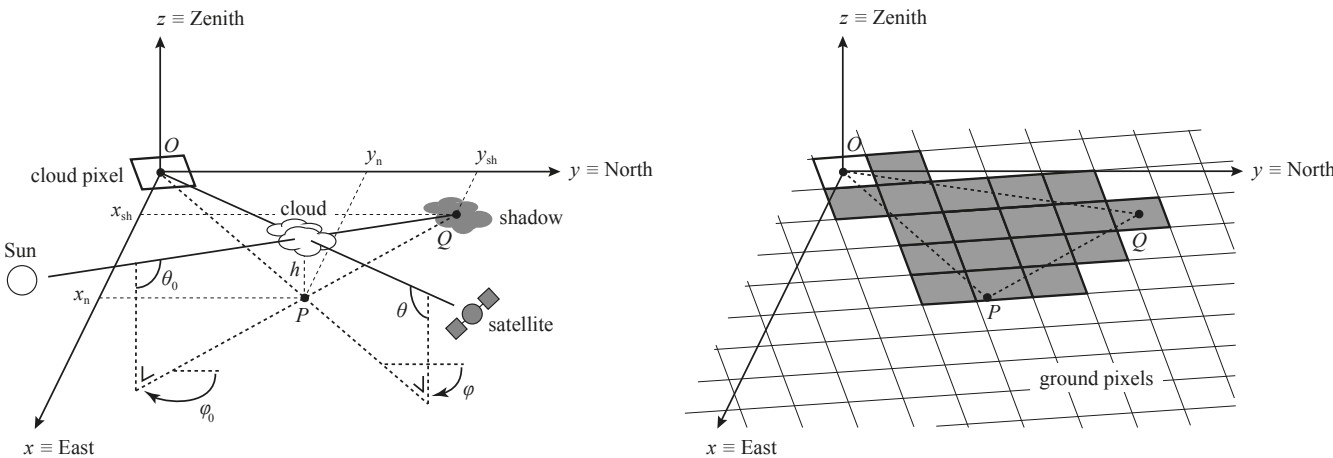

**Figure 2.** Sketch of the cloud shadow geometry in the local reference frame at the Earth's surface. The cloud as observed by the satellite is located at point $O$, resulting in a TROPOMI cloud pixel at $O$ (indicated by the white quadrilateral), while the actual cloud is located at height $h$ above point $P$. The shadow is located at point $Q$.

**Figure 3.** Sketch of the PCSF flagging of the TROPOMI ground pixels in the local reference frame at the Earth's surface. The PCSF pixels are indicated in grey and the cloud pixel is indicated by the white quadrilateral. Points $O$, $P$. and $Q$ correspond to points $O$, $P$ and $Q$ in Fig. 2.

In Eq. (1), $h_{\mathrm{c}}$ is the FRESCO cloud height (Koelemeijer et al., 2001; Wang et al., 2008), which is an approximation of the true height of the cloud's centroid w.r.t. the WGS84 Earth reference ellipsoid. Because, for the PCSF, we search for the maximum potential shadow extent, we have introduced the safety margin $C$ which raises the cloud proportional to $h_{\mathrm{c}} - h_{\mathrm{sfc}}$. We set $C = 0.5$, for which the number of false negative shadow detections (see Sect. 4) resulting from underestimated maximum potential shadow extents converged to a minimum.

If we assume that the center of the cloud pixel is the projection of the cloud's centroid in the viewing direction onto the Earth's surface, $x_{\mathrm{n}}$ and $y_{\mathrm{n}}$ can be computed as (see also Luo et al., 2008):

$$x_{\mathrm{n}} = h \cdot \tan\theta \sin\varphi, \tag{2}$$

$$y_{\mathrm{n}} = h \cdot \tan\theta \cos\varphi. \tag{3}$$

The location of point $Q$ in the cloud shadow on the Earth's surface, $(x_{\mathrm{sh}}, y_{\mathrm{sh}})$, then follows from:

$$x_{\mathrm{sh}} = x_{\mathrm{n}} - h \cdot \tan\theta_0 \sin\varphi_0, \tag{4}$$

$$y_{\mathrm{sh}} = y_{\mathrm{n}} - h \cdot \tan\theta_0 \cos\varphi_0. \tag{5}$$

Finding the geodetic latitude, $\delta_{\mathrm{sh}}$, and longitude, $\vartheta_{\mathrm{sh}}$, of $Q$ is an example of the direct geodetic problem for which the solution involves an iterative procedure (Vincenty, 1975). However, because of the small distances in the cloud shadow geometry relative





to the Earth's radii of curvature, $\delta_{\rm sh}$ and $\vartheta_{\rm sh}$ can accurately be approximated by differential northing and easting formulae:

$$\delta_{\rm sh} \approx \delta_{\rm c} + \frac{y_{\rm sh}}{M + h_{\rm sfc}}, \tag{6}$$

$$\vartheta_{\rm sh} \approx \vartheta_{\rm c} + \frac{x_{\rm sh}}{(N + h_{\rm sfc})\cos\delta_{\rm c}}, \tag{7}$$

where $M$ and $N$ are the Earth's ellipsoidal meridian radius of curvature and radius of curvature in the prime vertical, respectively, which both vary with latitude $\delta_c$ (see e.g. Torge and Müller, 2012).

The center of the cloud pixel may not coincide with the projection of the cloud's centroid in the viewing direction onto the Earth's surface, as was assumed in Eqs. (2) and (3). This is particularly true, for example, when small clouds in the pixel are located near the pixel edges or corners, or when the edge of a large cloud deck traverses the pixel. Moreover, the unknown true horizontal and vertical cloud extents are projected inside but near the edges of the cloud pixel.[1] Therefore, we repeat the computation of point $Q$ four times, now placing the *corners* of the cloud pixel in the origin of the reference frame (not shown in Fig. 2).

### 2.1.2 Raising the PCSF

In the second step of the PCSF determination, we select the area in which PCSFs are to be raised, based on the calculated points $P$ and $Q$. As illustrated in Fig. 3, we flag all the cloud-free ground pixels within or intersected by the triangle $OPQ$.

All cloud-free ground pixels intersected by line segment $OQ$ are flagged for two reasons. First, $OQ$ is the projection in the viewing direction onto the Earth's surface of a line segment, between the cloud's centroid and point $Q$, where the shadowed atmosphere is located (e.g., an optically thick atmosphere may lead to short shadows, cast on the atmospheric layers, projected onto the surface close to point $O$). Secondly, a possible overestimation of $h$ implies an actual cloud's nadir projection closer to $O$ (along line $OP$) which, with an unchanged illumination geometry, results in a shadow location between $O$ and $Q$ on line segment $OQ$.

All cloud-free ground pixels intersected by line segment $PQ$ are flagged because the vertical cloud extent below the cloud's centroid is unknown. Although the vertical cloud extent of an isolated cloud may result in an adjacent cloud pixel, the vertical extent *below* the cloud's centroid may be invisible from space if neighboring clouds cover the volume below the cloud's centroid. For that reason, line segment $PQ$ represents the potential shadow locations of a hypothetical cloud extending from the cloud's centroid to the surface.

All cloud-free ground pixels within or intersected by triangle $OPQ$ are flagged, because combinations of the aforementioned situations may occur. For similar reasons as mentioned in Sect. 2.1.1, we repeat the flagging four times for the triangles $OPQ$ where $O$ is placed in the corners of the cloud pixel (not shown in Fig. 3).

---

[1]An even larger horizontal or vertical cloud extent would be part of an adjacent cloud pixel.





## 2.2 Actual cloud shadow flag (ACSF)

In this section, the computation of the ACSF is explained. The ACSFs indicate the pixels that are confidently affected by cloud shadows. They are a subset of the PCSFs, and are intended to be useful for selecting pixels for the scientific analysis of cloud shadows. The number of false positive shadow detections in the ACSF should therefore be minimized (see Sect. 4).

The ACSF is determined in two steps. First, we apply a Rayleigh scattering correction to the measured reflectance at the top of the atmosphere for the PCSF pixels. Then, we compare this corrected reflectance to the expected surface reflectance from climatological observations by TROPOMI, revealing the actual shadowed pixels. The ACSF determination is based on the darkness of the shadowed pixels with respect to non-shadowed pixels, which is most apparent at the wavelength where the surface reflectance is strongest. Both steps are explained in more detail below.

### 2.2.1 Lambertian-equivalent reflectivity of the scene (SCNLER)

The spectral Earth's reflectance at the top of the atmosphere (TOA) as measured by a satellite is defined as

$$R^{\mathrm{meas}}(\mu, \mu_0, \varphi, \varphi_0, \lambda) = \frac{\pi I(\mu, \mu_0, \varphi, \varphi_0, \lambda)}{\mu_0 E_0(\lambda)}, \tag{8}$$

where $I$ is the radiance reflected by the atmosphere-surface system in W m$^{-2}$sr$^{-1}$nm$^{-1}$, $E_0$ is the extraterrestrial solar irradiance perpendicular to the beam in W m$^{-2}$nm$^{-1}$ and $\mu_0 = \cos\theta_0$. $I$ and $E_0$ depend on wavelength $\lambda$ in nm, and $I$ additionally depends on $\mu = \cos\theta$, $\mu_0$, $\varphi$ and $\varphi_0$.

First, we calculate the albedo of the surface, $A_{\mathrm{s}}$, needed to match a modeled TOA reflectance, $R^{\mathrm{model}}$, to the measured TOA reflectance, $R^{\mathrm{meas}}$. The model assumes a Lambertian (i.e., depolarizing and isotropic reflecting) surface below a cloud-free and aerosol-free atmosphere, such that the modeled TOA reflectance can be expressed as (Chandrasekhar, 1960):

$$R^{\mathrm{model}}(\mu, \mu_0, \varphi - \varphi_0, \lambda) = R^0(\mu, \mu_0, \varphi - \varphi_0, \lambda) + \frac{A_{\mathrm{s}}(\lambda)T(\mu, \mu_0, \lambda)}{1 - A_{\mathrm{s}}(\lambda)s^*(\lambda)}. \tag{9}$$

The first term at the right-hand side of Eq. (9), $R^0$, is the so-called path reflectance, which is the modeled TOA reflectance of the atmosphere bounded below by a black surface. The second term is the modeled surface contribution to the TOA reflectance, where $A_{\mathrm{s}}$ is the albedo of the Lambertian surface, $T$ is the total transmittance of the atmosphere for illumination from above and below, and $s^*$ is the spherical albedo of the atmosphere for illumination from below. Quantities $R^0$, $T$ and $s^*$ of the cloud-free and aerosol-free atmosphere-surface model were prepared with the 'Doubling-Adding KNMI' (DAK) radiative transfer code (de Haan et al., 1987; Stammes, 2001), version 3.2.0, taking into account single and multiple Rayleigh scattering of sunlight by molecules in a pseudo-spherical atmosphere, including polarization. Absorption by $O_3$, $NO_2$, $O_2$, $H_2O$ and the $O_2$-$O_2$ collision complex was taken into account. For more details about the computation of the quantities in Eq. (9), we refer to Tilstra (2021).

The albedo $A_{\mathrm{s}}$ for which $R^{\mathrm{model}}(\lambda) = R^{\mathrm{meas}}(\lambda)$ holds is in this paper indicated by $A_{\mathrm{scene}}$. The expression for $A_{\mathrm{scene}}$ follows from Eq. (9) (see e.g. Tilstra et al., 2017):

$$A_{\mathrm{scene}}(\lambda) = \frac{R^{\mathrm{meas}}(\lambda) - R^0(\lambda)}{T(\lambda) + s^*(\lambda)(R^{\mathrm{meas}}(\lambda) - R^0(\lambda))}, \tag{10}$$





where the notation for the dependency on $\mu$, $\mu_0$, $\varphi$ and $\varphi_0$ is omitted. We compute $A_{\text{scene}}$ for $\lambda = 402, 416, 425, 440, 463, 494,$ 670, 685, 696.97, 712.7, 747, 758 and 772 nm, and co-register the results at NIR wavelengths to the Level 2 UVIS ground pixel grid. The values of $A_{\text{scene}}$ can be interpreted as the TOA reflectances of the scene corrected for molecular Rayleigh scattering.

They are in fact scene albedos, because they include non-Lambertian surface, aerosol, cloud and shadow effects. Therefore, in what follows, we refer to $A_{\text{scene}}$ as the Lambertian-equivalent reflectivity of the scene (SCNLER). Only in the absence of non-Lambertian effects, $A_{\text{scene}}$ is independent of $\mu$, $\mu_0$, $\varphi$ and $\varphi_0$ and approximates the true surface albedo.

### 2.2.2 Directionally dependent Lambertian-equivalent reflectivity (DLER) climatology

In the second step of the ACSF determination, the SCNLER of the PCSF pixels is compared to climatological observations at

the same coordinates and time of the year. For the climatological observations, we use the directionally dependent Lambertian-equivalent reflectivity (DLER) data[2] version 0.6 generated with TROPOMI observations of the SCNLER since the start of TROPOMI's operational phase in May 2018. The DLER is available on a global 0.125° by 0.125° resolution latitude-longitude grid for each calendar month at 21 one-nm wide wavelength bins between 328 and 2314 nm (Tilstra, 2021). We linearly interpolate the DLER data to the TROPOMI Level 2 UVIS ground pixel grid and measurement times. Unless stated otherwise,

the wavelength bins we use are centered at 402, 416, 425, 440, 463, 494, 670, 685, 696.97, 712.7, 747, 758 and 772 nm.

In the DLER algorithm, the 10% lowest SCNLER measurements in the seasonal grid cell were used, and measurements containing aerosols or clouds were excluded (see Tilstra, 2021). The DLER can generally be considered shadow-free. The DLER takes into account the viewing zenith angle dependence of the SCNLER caused by non-Lambertian surface reflectance. The DLER is a more accurate estimate of the expected aerosol-, cloud- and shadow-free SCNLER than the traditionally used

LER (without viewing zenith angle dependence). The viewing zenith angle dependence of the DLER is only taken into account over land surfaces. Over water surfaces, DLER = LER. For more details about the DLER theory, we refer to Tilstra et al. (2021).

### 2.2.3 Raising the ACSF

In order to select the pixels for which an ACSF is to be raised, we define the SCNLER-DLER contrast parameter $\Gamma$:

$$\Gamma(\lambda) = \frac{A_{\text{scene}}(\lambda) - A_{\text{DLER}}(\lambda)}{A_{\text{DLER}}(\lambda)} \times 100\%.$$
(11)

The division by $A_{\text{DLER}}$ in Eq. (11) allows us to search for a $A_{\text{DLER}}$-independent ACSF threshold for $\Gamma$, that is, a single threshold that can be used for both dark and bright surface types. Because of the division by the DLER, $\Gamma$ is more stable (i.e., less susceptible to potential offset errors in the DLER) when the DLER is high. For each PCSF pixel, we compute the wavelength for shadow detection, $\lambda_{\text{max}}$, at which the pixel's DLER is maximum:

$$\lambda_{\text{max}} = \underset{\lambda}{\text{argmax}}\ A_{\text{DLER}}(\lambda).$$
(12)

---

[2]See https://www.temis.nl/surface/albedo/tropomi_ler.php, visited on 23 October 2021.



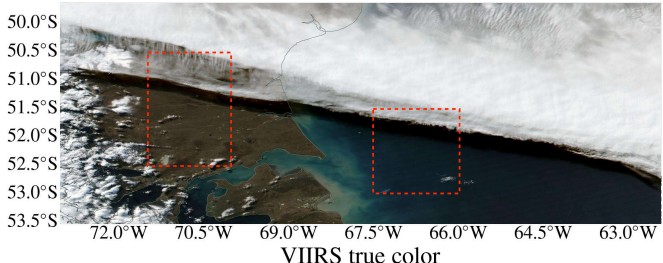

**Figure 4.** VIIRS-NPP true color image of Southern Chile and Argentina on 3 August 2019. The land and water regions belonging to the spectra of Fig. 5 are indicated by red dashed boxes.

We raise an ACSF at PCSF pixels for which

$$\Gamma(\lambda_{\max}) < q, \tag{13}$$

where $q$ is the contrast threshold. We set $q = -15\%$, yielding the highest actual shadow detection score in the validation (see Sect. 4).

Here, we demonstrate the behavior of the variables used in Eqs. (11) to (13) with an example measurement. Figure 4 is a true color image made by the Visible Infrared Imager Radiometer Suite (VIIRS) instrument on board the Suomi National Polar-orbiting Partnership (NPP) satellite, on 3 August 2019 above Southern Chile and Argentina. Suomi NPP orbits in constellation with S5P: the measurement time intervals of TROPOMI and VIIRS were 19:00-19:01 UTC and 18:57-18:58 UTC, respectively. A specific land region (52.5°-50.5°S latitude and 71.5°-70°W longitude) and water region (53°-51.5°S latitude

and 67.5°-66°W longitude) are indicated by red dashed boxes. The main surface types in those regions are steppe and ocean, respectively. Figures 5a and 5b show the spectral behavior of the mean and 1-$\sigma$ of SCNLER measurements affected by shadow ($A_{\mathrm{scene}}$ shadow) and not affected by shadows ($A_{\mathrm{scene}}$ no shadow) of cloud-free TROPOMI pixels in the land and water region, respectively. We used the PCSF to remove shadow pixels and the ACSF to select shadow pixels. Also shown are the mean and 1-$\sigma$ of the DLER interpolated on the TROPOMI Level 2 UVIS grid.

Figure 5a shows that over land (steppe), the DLER and the cloud- and shadow-free SCNLER follow a typical surface reflectivity spectrum for grasslands (cf. Fig. 7 of Tilstra et al., 2017): they increase with increasing wavelength, and include a subtle signature of the so-called 'red edge' (i.e., the sudden surface albedo increase at $\lambda \sim 700$ nm caused by vegetation). Over ocean, the DLER and cloud- and shadow-free SCNLER follow a typical surface reflectivity spectrum for ocean water: they increase with decreasing wavelength, and peak at $\lambda \sim 400$ nm where the peak significance depends on the water constituents

(see also e.g. Morel and Maritorena, 2001). The mean value of $A_{\mathrm{scene}}$ affected by shadow is smaller than the DLER and cloud- and shadow-free $A_{\mathrm{scene}}$ at all wavelengths, for both the land and water region. The shadow signature in the difference $A_{\mathrm{scene}} - A_{\mathrm{DLER}}$ is most evident at the wavelength where the DLER is highest. The Rayleigh scattering correction results in negative $A_{\mathrm{scene}}$ for part of the shadowed pixels. Above land, a slight increase of the shadowed $A_{\mathrm{scene}}$ can still be observed with increasing wavelength, but above ocean, the water albedo increase in the UV cannot be observed anymore. Note that the mean





**(a)**          **(b)**

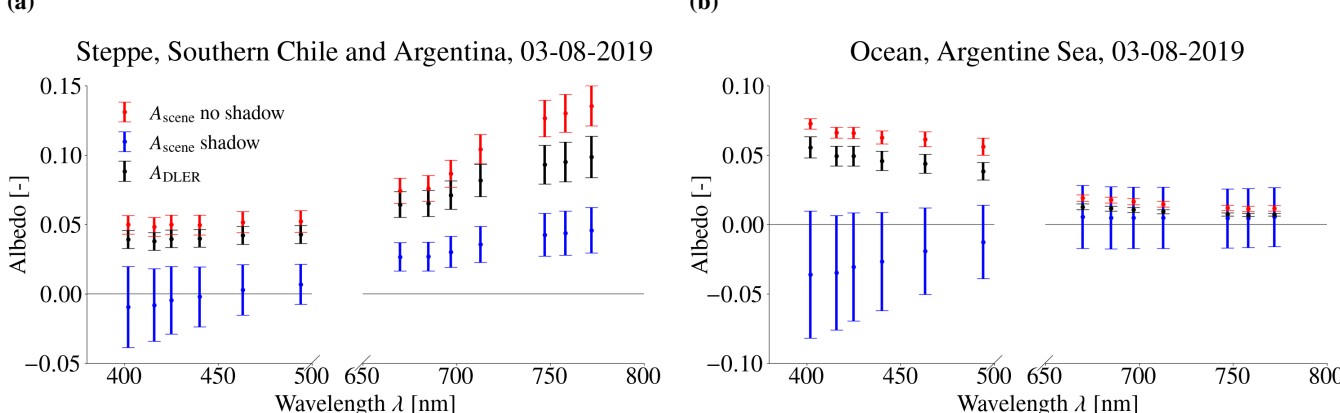

**Figure 5.** Spectra of the mean and 1-$\sigma$ of the Lambertian-equivalent scene reflectivity (SCNLER) measured by TROPOMI at Southern Chile and Argentina on 3 August 2019, for the steppe region within 52.5°-50.5°S latitude and 71.5°-70°W longitude (Fig. 5a) and for the ocean region within 53°-51.5°S latitude and 67.5°-66°W longitude (Fig. 5b). Here, all measurements are cloud-free. The measurements affected by shadow (i.e., with ACSF) are presented in blue, and the shadow-free measurements (i.e., without PCSF) are presented in red. The additional black spectra are of the mean and 1-$\sigma$ of the directionally dependent climatological Lambertian-equivalent reflectivity (DLER) at the TROPOMI ground pixels in the particular regions.

DLER is consistently smaller than the mean cloud- and shadow-free SCNLER measured at all wavelengths, which is expected since the DLER at a certain location was generated with the 10% lowest SCNLER values at that location.

Figure 6a shows $\lambda_{\mathrm{max}}$ on the TROPOMI Level 2 UVIS ground pixel grid for this measurement example. As expected from Fig. 5, $\lambda_{\mathrm{max}} = 772$ nm for the majority of the land covered pixels and $\lambda_{\mathrm{max}} = 402$ nm for the majority of the water covered pixels. In shallow water regions near the coast line, however, $\lambda_{\mathrm{max}} = 494$ nm, while in some land coast regions we find

$\lambda_{\mathrm{max}} = 670$ nm. Indeed, employing $\lambda_{\mathrm{max}}$, the usage of surface type classification flags is avoided (see Romahn et al., 2021, for an example of surface type classification flags usage). That is, $\lambda_{\mathrm{max}}$ does not rely upon assumptions made in a surface type classification product, and will also give the most suitable wavelength for shadow detection when mixed and/or rare surface types are present within the pixel.

Figures 6b, 6c and 6d show $A_{\mathrm{scene}}$, $A_{\mathrm{DLER}}$ and $\Gamma$, respectively, at $\lambda_{\mathrm{max}}$. Cloud- and shadow-free pixels yield $\Gamma(\lambda_{\mathrm{max}}) \sim 0\%$ or

slightly positive (up to $\sim 50\%$), because the DLER is generated with the 10% lowest SCNLER values in the particular calendar month that passed an aerosol- and cloud screening. The clouds at latitudes larger than 52.5° increase $A_{\mathrm{scene}}$ significantly relative to $A_{\mathrm{DLER}}$, which results in $\Gamma(\lambda_{\mathrm{max}}) > 50\%$. Pixels affected by true shadows show a significantly decreased $A_{\mathrm{scene}}$ relative to $A_{\mathrm{DLER}}$, which is most apparent for the elongated cloud shadow along the edge of the cloud deck.



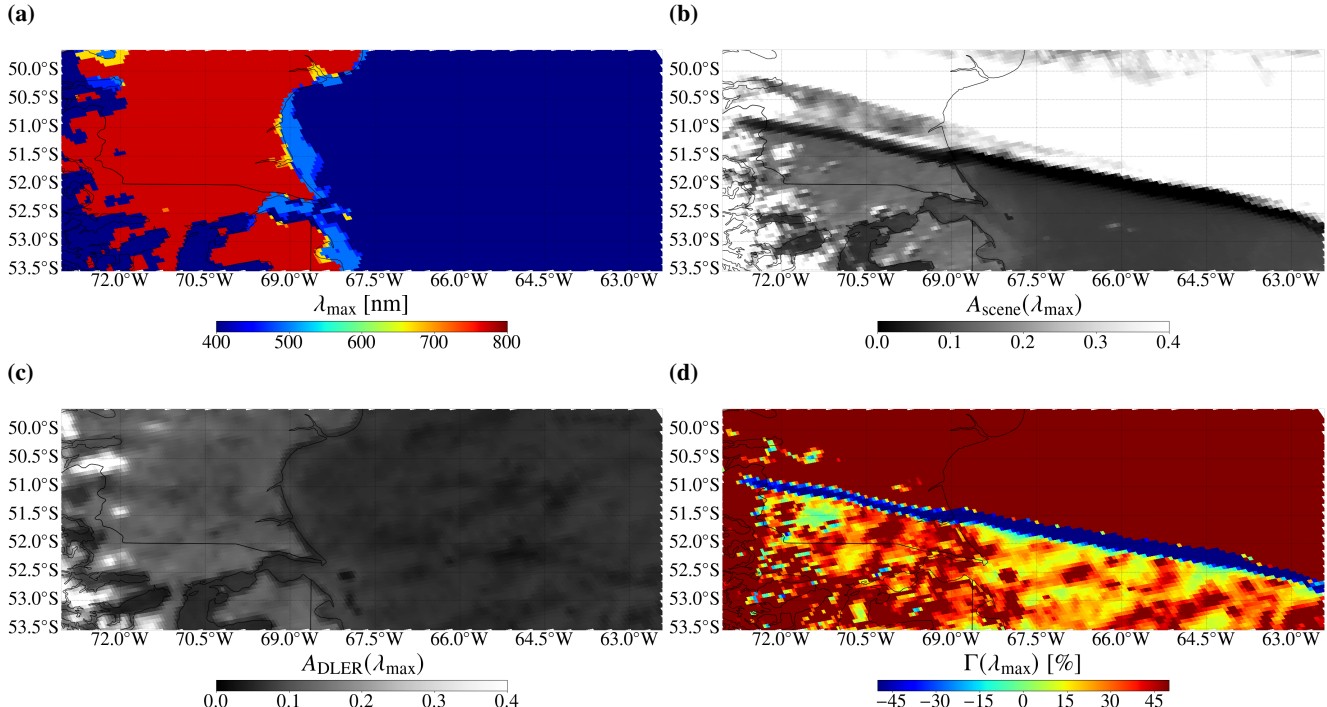

**Figure 6.** The wavelength at which DLER is maximum $\lambda_{max}$ (Fig. 6a), the SCNLER at $\lambda_{max}$ (Fig. 6b), the DLER at $\lambda_{max}$ (Fig. 6c), and contrast parameter $\Gamma$ at $\lambda_{max}$ (Fig. 6d), for Southern Chile and Argentina on 3 August 2019.

## 3 Results

Here, we discuss the potential and actual cloud shadow flag results for three case studies with different cloud and surface types: the cloud deck example above steppe and ocean surfaces introduced above 2.2 (Sect. 3.1), an example with patchy clouds above grass and forest surfaces (Sect. 3.2) and an example of a relatively large area above the Sahara desert containing thin cirrus clouds (Sect. 3.3).

### 3.1 Southern Chile and Argentina, 3 August 2019

Figures 7a and 7b show (in blue) the TROPOMI Level 2 UVIS ground pixels with raised PCSFs and ACSFs, respectively, for the cloud shadow example on 3 August 2019 at Southern Chile and Argentina.

Figure 7a shows that the PCSFs indicate the presence of an elongated cloud shadow southward of the edge of the cloud deck longitudinally traversing the scene from $\sim 51°$S to $\sim 52.5°$S latitude. The southward shadow is expected because in this example the Sun is located in the Northwest ($\varphi_0$ ranges from -29.3° in the West to -41.7° in the East). The Sun is located

relatively low in the sky because of the local winter season ($\theta_0$ ranges from 72.2° in the Northwest to 79.7° in the Southeast), which is geometrically beneficial for the existence of long shadows (see Eq. (5)). The latitudinal extent of the elongated shadow is relatively large compared to the shadows of the isolated small clouds found at latitudes southward of $51.5°$S. This variation





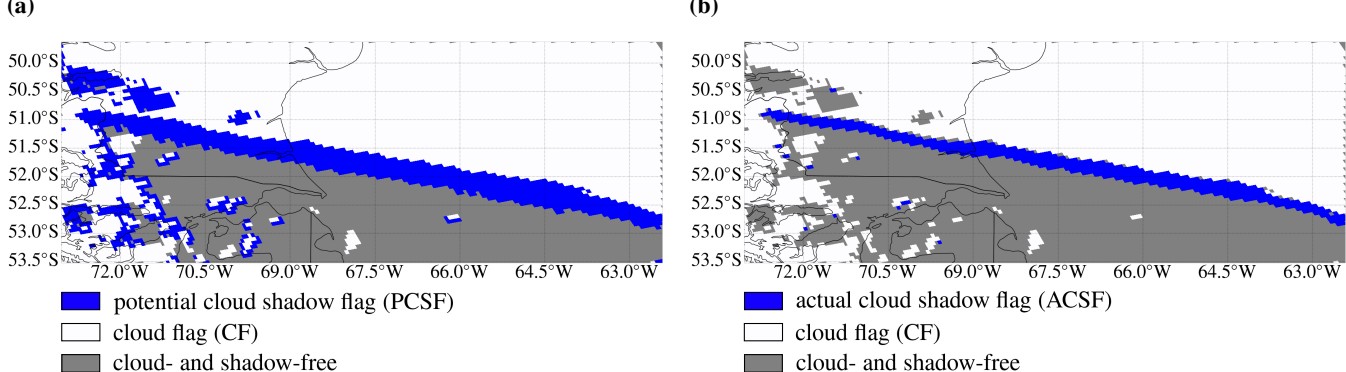

**Figure 7.** The TROPOMI Level 2 UVIS ground pixels for Southern Chile and Argentina on 3 August 2019 with raised PCSFs (Fig. 7a) and with raised ACSFs (Fig. 7b), indicated in blue. The white pixels are cloud pixels and grey pixels do not contain a raised cloud or shadow flag.

of shadow extent can be explained by the difference in cloud height: $h \sim 15$ km for the cloud deck, while $h \sim 1$ km for the isolated small clouds. The cloud deck shadow extent is larger than expected from visual inspection of the true color image (Fig.

4), which is caused by the cloud height safety margin $C$ that we included in Eq. (1).

Figure 7b shows that the latitudinal extent of the cloud deck shadow detected with the ACSF is a more realistic approximation of the latitudinal cloud deck shadow extent observed in the true color image of Fig. 4. Only few shadows of the small isolated clouds are detected with the ACSF. Note that part of the small isolated clouds are in fact false positive cloud detections in the cloud product caused by a temporarily bright surface. This can readily be concluded by comparing Fig. 7b to Fig. 4. For

example, the water constituents along the coast between 53°S and 53.5°S latitude, but also the snowy mountains westward of 71°W, are falsely interpreted as clouds. These false cloud shadow detections in the PCSF are correctly filtered out by the threshold for $\Gamma(\lambda_{\max})$ (Eq. (13)), and are therefore not part of the ACSF. Indeed, the performance of the shadow detection algorithm depends on the quality of the input cloud and DLER products. The gaps in the cloud deck between $51°$S and $50°$S are caused by undefined cloud fractions in the cloud product, but again, the false PCSF shadow detections within those gaps

are (except for 2 pixels) correctly removed from the ACSF.

### 3.2 The Netherlands and Germany, 18 November 2018

Figures 8a and 8c show the true color image and the TROPOMI Level 2 UVIS ground pixels with raised PCSFs, respectively, for an example on 18 November 2018 above the Netherlands and Germany. TROPOMI orbits northwestward, and the viewing geometry is southwestward: $\theta$ ranges from 8.8° in the Northeast to 54.3° in the Southwest. The Sun is located in the South ($\varphi_0$

ranges from -180.0° in the West to -165.7° in the East) and the solar zenith angle $\theta_0$ ranges from 65.8° in the South to 76.8° in the North.

With the PCSF, long potential cloud shadows are found extending towards the Northeast. Here, all clouds that produce shadows are relatively high ($h \sim 10$ km or higher). Note that at the location of the small isolated clouds above the sea at $\sim$ 54°N latitude and 4.5-6°E longitude, the Sun is almost directly located in the South. The eastward component of the potential





**(a)**

**(b)**

**(c)**

**(d)**

**Figure 8.** VIIRS-NPP true color image (Fig. 8a), SCNLER-DLER contrast parameter $\Gamma$ at $\lambda_{max}$ measured by TROPOMI (Fig. 8b), TROPOMI Level 2 UVIS ground pixels with raised PCSFs (Fig. 8c), and with raised ACSFs (Fig. 8d), for the Netherlands and Germany on 18 November 2018. In Figs. 8c and 8d, white pixels are cloud pixels and grey pixels do not contain a raised cloud or shadow flag.

shadow at these longitudes is caused by the parallax effect (cf. Fig. 2): the southwestward looking instrument projects the cloud as a cloud pixel onto the surface southwestward from the cloud's actual nadir location. Although the path from the cloud's nadir location to the actual shadow is strictly northward, the path from the cloud *pixel* to the actual shadow is northeastward.

     The majority of the cloud shadows in this example are found above land surface. The main land surface types in this part of Europe are grassland and forest, with in general a higher vegetation density than for the steppe land in the example shown

in Sect. 3.1. Consequently, the red edge is more pronounced in this example resulting in a stronger surface reflectance in the





near-infrared. Hence, we find that $\lambda_{\max}$ = 772 nm for all pixels over land. The relatively high surface reflectance in the near-infrared results in a clear shadow signature in $\Gamma(\lambda_{\max})$ (see Fig. 8b): in the cloud- and shadow-free regions $\Gamma(\lambda_{\max})$ equals 0 or is slightly positive, while strong negative $\Gamma(\lambda_{\max})$ values are confined to cloud shadows (cf. Fig. 8a).

Figure 8d shows the TROPOMI Level 2 UVIS ground pixels with raised PCSFs for this example. Comparing the shadows
detected with the ACSF to the true color image of Fig. 8a shows that ACSF shadows are detected where they can be expected. The small high isolated clouds above the sea do not produce dark enough shadows for an ACSF to be raised, similar to the small high isolated clouds above land at ∼49.5°N latitude and ∼10.5°-11°E longitude.

### 3.3   Sahara desert, 18 January 2021

Figure 9 is equivalent to Fig. 8, but then for an example above the Sahara desert on 18 January 2021. The area of this example
covers most of the orbit swath of TROPOMI traveling north-northwestward: $\theta$ ranges from 66.5° in the West-southwest to -58.1° in the East-northeast. Although the latitudes in this example are relatively small, the local winter season causes the Sun not to be located directly overhead ($\theta_0$ ranges from 28.4° in the South to 59.3° in the North, and $\varphi_0$ ranges from -178.7° in the West to -140.3° in the East).

With the PCSF, northward shadows of longitudinally elongated cirrostratus clouds between 25°N and 28°N latitude, and of
cirrocumulus clouds between 13°N and 22.5°N latitude, are detected. For both cloud types, $h > 10$ km. The vertical location of the detected foggy patch at 15°-17.5°N latitude and 14°-19°E longitude is just above the surface, hence the absence of the potential shadow (see Fig. 9c). This example is a clear demonstration of the parallax effect: on the west side of the area, TROPOMI looks westward projecting the clouds as cloud pixels onto the surface 'too far' westward, resulting in an *eastward* component of the potential shadow locations w.r.t. the cloud pixels. Similarly, on the east side of the area, TROPOMI looks
eastward and potential shadows tend to be located *westward* of the cloud pixels.

With the ACSF, the detected shadows are a more accurate approximation of the shadows observed in the true color image (cf. Figs. 9d and 9a). The most distinctive shadow signature in the true color image, which is the northward shadow of the longitudinally elongated cirrostratus cloud between 25°N and 27°N latitude, is indeed also detected by the ACSF. Although, geometrically, many other clouds in this example are high enough to produce potential cloud shadows, some of those clouds are
too small and/or too thin to produce actual cloud shadows. This can be seen in Fig. 9b: for example, the cirrocumulus clouds near 13.5°N latitude and 4.5°E longitude are not able to decrease $\Gamma(\lambda_{\max})$ significantly enough for an ACSF to be raised.

The spectral reflectance of desert surface does not contain a red edge, but is relatively strong already at $\lambda < 700$ nm and further increases with increasing wavelength (see e.g. Fig. 7 of Tilstra et al., 2017). We find that for almost all pixels in this example, $\lambda_{\max}$ = 772 nm. Comparing Fig. 9b with the false color image of Fig. 9a shows that strong negative $\Gamma(\lambda_{\max})$ are strictly
confined to cloud shadows (except for a few pixels near 12.7°N latitude and 17.8°E longitude). The cloud- and shadow-free area yield $\Gamma(\lambda_{\max}) \sim 0$ or slightly positive. That is, dark surface features in the Sahara desert are not falsely detected as cloud shadows.





**Figure 9.** Similar to Fig. 8, but for the Sahara desert on 18 January 2021.



## 4 Validation

In this section, we validate DARCLOS by comparing the computed PCSFs and ACSFs to the shadows visually found at similar

locations and time in VIIRS-NPP true color images. For the visual inspection of the true color images, we have developed an interactive Python tool which plots the TROPOMI Level 2 UVIS grid on top of the true color image. The software allows for the manual selection and de-selection of TROPOMI pixels containing VIIRS shadows by clicking on the image, after which the row and scanline numbers of the selected TROPOMI pixels are saved.

Figure 11a shows a VIIRS-NPP true color image of cloud shadows found at the Taklamakan desert at Xinjiang, China, on 22

December 2019. The red lines represent the TROPOMI Level 2 UVIS grid, and the blue crosses indicate the TROPOMI pixels with a raised ACSF. If, also, a VIIRS shadow is visually found at the TROPOMI pixel with a raised shadow flag, we speak of a true positive (TP) shadow detection. Similarly, we register the false positive (FP), false negative (FN), and true negative (TN) shadow detections (see Fig. 10).

The overestimation of the VIIRS shadow by DARCLOS can be expressed by the commission error, $\epsilon_C$ (see also e.g. Candra

et al., 2016):

$$\epsilon_C = \frac{N_{FP}}{N_{TP} + N_{FP}}, \tag{14}$$

where $N_{FP}$ and $N_{FN}$ are the number of false positive detections and the number of false negative detections, respectively. The underestimation of the VIIRS shadow by DARCLOS can be expressed by the omission error, $\epsilon_O$:

$$\epsilon_O = \frac{N_{FN}}{N_{TP} + N_{FN}}, \tag{15}$$

where $N_{FN}$ are the number of false negative detections. For the definition of the VIIRS shadows, we distinguish between TROPOMI pixels that are totally shadowed (with geometrical shadow fractions $\gtrsim 0.75$), and partly shadowed (with geometrical shadow fractions $\gtrsim 0$ and $\lesssim 0.75$). For the computation of $\epsilon_O$, we use only the totally shadowed pixels, while for the computation of $\epsilon_C$, we use both the totally and partly shadowed pixels. That is, we consider the underestimation of the totally shadowed pixels, and the overestimation of the totally and partly shadowed pixels, to be erroneous. The overall performance

of the algorithm can be assessed with the $F_1$ score which combines $\epsilon_C$ and $\epsilon_O$ as follows (see e.g. Fernández et al., 2018):

|  | VIIRS shadow | VIIRS no shadow |
|---|---|---|
| TROPOMI shadow | True positive (TP) | False positive (FP) |
| TROPOMI no shadow | False negative (FN) | True negative (TN) |

**Figure 10.** Confusion matrix of the shadow detection on the TROPOMI Level 2 UVIS grid. TROPOMI shadows are detected with the PCSF or ACSF of DARCLOS. VIIRS shadows are manually determined by visual inspection of VIIRS true color images.





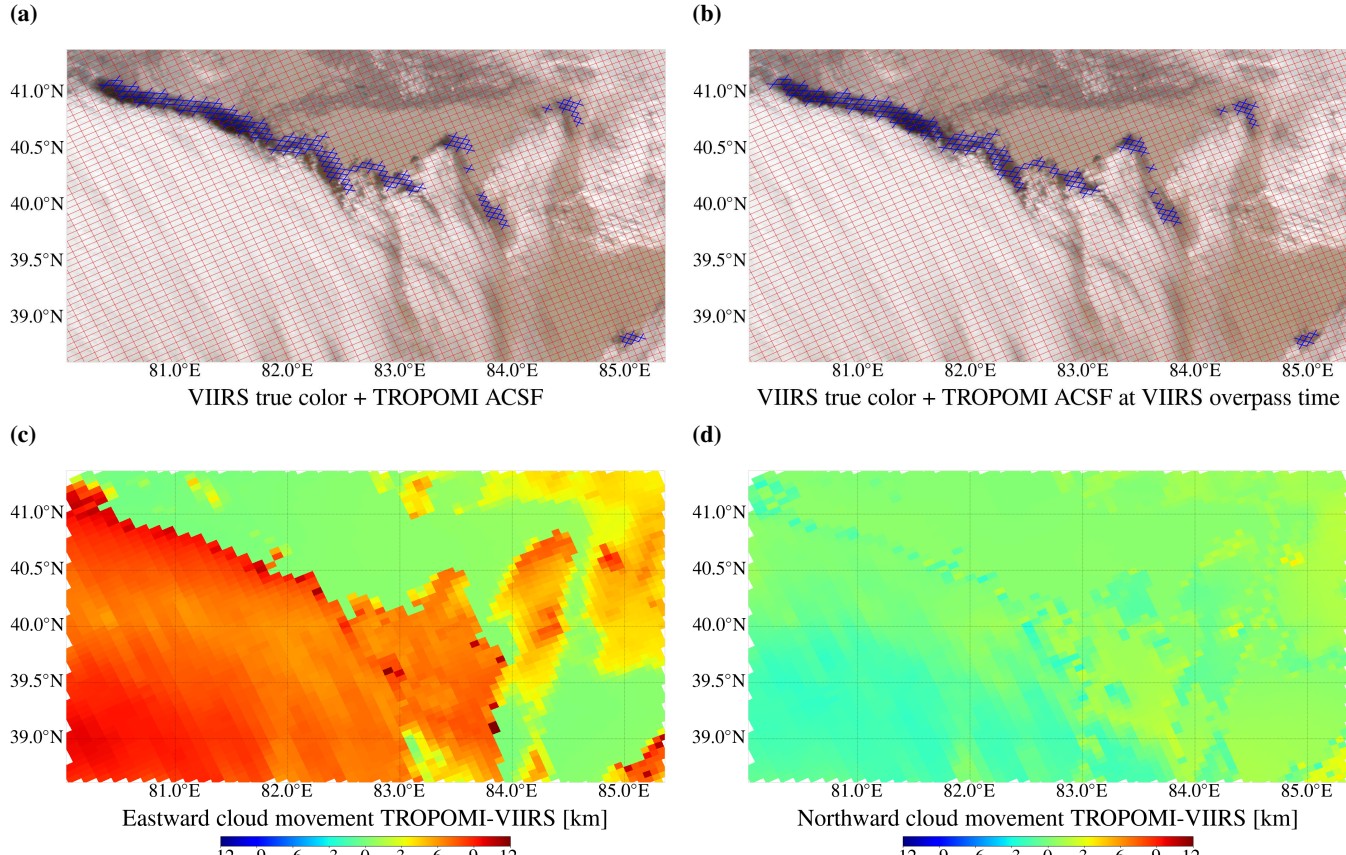

**Figure 11.** VIIRS-NPP true color image of the Taklamakan desert at Xinjiang, China, 22 December 2019, with the TROPOMI Level 2 UVIS grid plotted on top (in red) and the detected ACSF by DARCLOS (blue crosses), uncorrected (Fig. 11a) and at the VIIRS measurement times (Fig. 11b). Figures 11c and 11d show, respectively, the eastward and northward atmospheric movement at the cloud height during the TROPOMI-VIIRS measurement time difference, computed using ERA5 reanalysis wind data.

$$F_1 = \frac{2(1 - \epsilon_C)(1 - \epsilon_O)}{(2 - \epsilon_C - \epsilon_O)}. \tag{16}$$

In the hypothetical case of a perfect shadow detection, we would obtain $\epsilon_O = 0$, $\epsilon_C = 0$ and $F_1$ score = 1. For the ACSF in Fig. 11a, the results are $\epsilon_O = 0.27$, $\epsilon_C = 0.21$ and $F_1$ score = 0.76.

It can be observed that the TROPOMI pixels with a raised ACSF in Fig. 11a are consistently located slightly eastwards of 385 the shadows found in the VIIRS true color image. The eastward shift can be explained by the motion of the clouds during the measurement time difference of TROPOMI and VIIRS. In the example of Fig. 11, the VIIRS measurements were on average taken 4.33 minutes ahead of the TROPOMI measurements, with a 1-$\sigma$ of 0.07 minutes[3]. We interpolate ERA5 data (Hersbach

---

[3]TROPOMI-VIIRS measurement times differences were taken from the S5P-NPP cloud product, which is the cloud product of VIIRS regridded to the TROPOMI Level 2 grid (see Siddans, 2016).





| Example | Coordinates | Date | Orbit | Omission error PCSF | Omission error ACSF | Commission error ACSF | $F_1$ score ACSF |
|---|---|---|---|---|---|---|---|
| Southern Chile and Argentina | 53.528-49.626 °S 73.047-62.418 °W | 03-08-2019 | 9355 | **0.05** | 0.10 | 0.01 | **0.94** |
| The Netherlands and Germany | 49.004-54.991 °N 3.4119-12.5062 °E | 18-11-2018 | 5690 | **0.06** | 0.16 | 0.04 | **0.90** |
| Sahara desert, North Africa | 24.802-27.400 °N 3.506 - 12.011 °E | 18-01-2021 | 16927 | **0.14** | 0.18 | 0.13 | **0.84** |
| Taklamakan desert, China | 36.500-43.000 °N 76.000-88.000 °E | 22-12-2019 | 11348 | **0.02** | 0.08 | 0.02 | **0.95** |
| The Netherlands, Belgium and Luxembourg | 48.995-55.004 °N 2.000-8.000 °E | 09-10-2018 | 5123 | **0.05** | 0.20 | 0.07 | **0.86** |
| Taklamakan desert, China | 37.006-42.005 °N 80.005-88.007 °E | 21-12-2020 | 16527 | **0.10** | 0.13 | 0.11 | **0.88** |

**Table 1.** Results of the validation of the PCSF and ACSF by inspection of VIIRS-NPP true color images. The final results are shown in bold.

et al., 2018) of hourly eastward and northward wind speed components, provided at 37 vertical pressure levels on a 0.25° by 0.25° latitude-longitude grid, onto the FRESCO cloud pressure on the TROPOMI Level 2 UVIS grid (i.e., *without* manually raising the cloud height such as in Eq. (1)). The cloud deck in the South-west in Fig. 11 is relatively high (the cloud pressure is $\sim$ 400 hPa), where eastward wind speeds between 20 and 40 m/s are found. The eastward and northward cloud displacements are shown in Fig. 11c and 11d, respectively. The cloud displacements from $\sim$ 6 to $\sim$ 9 km are significant enough to shift some clouds, and hence some cloud shadows, at least one TROPOMI ground pixel in the eastward direction.[4]

In Fig. 11b we have corrected the locations of the TROPOMI cloud and cloud shadow pixels for the eastward and northward movement of the clouds during the TROPOMI-VIIRS measurement time difference. Note the much better agreement between the ACSF and the VIIRS shadows compared to Fig. 11a. Indeed, using the corrected ACSF, the errors decreased and the $F_1$ score increased: $\epsilon_O = 0.08$, $\epsilon_C = 0.02$ and $F_1$ score = 0.95. It should be noted that the validation may suffer from an imperfect correction of the cloud movement during the TROPOMI-VIIRS measurement time difference, because the cloud evolution is

---

[4]It should be noted that the cloud movement during the TROPOMI-VIIRS measurement time difference implies that the cloud flags (retrieved at the TROPOMI measurement time) cannot be replaced in DARCLOS by cloud flags from the S5P-NPP cloud product (retrieved at the VIIRS measurement time). Moreover, Fig. 11 is a general warning for all applications of the S5P-NPP cloud product which require a spatial cloud precision of about the size of a TROPOMI ground pixel.





ignored and because of the relatively coarse resolution of the wind product. Therefore, we expect the true shadow detection
performance at the TROPOMI measurement time to be even better than the performance presented with this validation.

The last 3 columns of Table 1 show the results for $\epsilon_O$, $\epsilon_C$ and the $F_1$ score of the ACSF for the three examples discussed
in this paper, and three additional examples (on the 22nd of December 2019, on the 9th of October 2018, and on the 21st of
December 2020) not shown in this paper. The $F_1$ score is 0.84 or higher for all examples. The $F_1$ score is highest (0.94 and
0.95 respectively) for Southern Chile and Argentina, 3 August 2019, and for the Taklamakan desert, 22 December 2019. The
shadows in those examples are caused by relatively large and thick cloud decks, and are therefore relatively distinctive. The
examples with the lowest $F_1$ scores (0.84 and 0.86 respectively) are the Sahara desert, 18 January 2021, and The Netherlands,
Belgium and Luxembourg, 9 October 2018. The shadows in those examples are caused by relatively thin and small clouds, and
are therefore relatively subtle. Subtle shadows lead to less distinctive shadow signatures in $\Gamma$, leading to more false negative
shadow detections and a higher $\epsilon_O$ of the ACSF. Also, thin and/or small clouds are sometimes not detected by the cloud product
because the cloud fraction is too low to raise a CF, resulting in false negative PCSFs and ACSFs. Moreover, we speculate
that thinner and/or smaller clouds are more likely to appear and disappear during the TROPOMI-VIIRS measurement time
difference, complicating the cloud movement correction and validation of these examples.

The fifth column of Table 1 shows $\epsilon_O$ of the PCSF. The value of $\epsilon_O$ of the PCSF is smaller than that of the ACSF, since the
shadow in the PCSF is, by definition, an overestimation of the actual shadow. Because the PCSF is intended to be useful for
excluding any cloud shadow contamination from TROPOMI Level 2 data, $\epsilon_O$ of the PCSF should be minimized. The value of
$\epsilon_O$ for all examples is 0.14 or lower. Also here, we attribute the nonzero $\epsilon_O$ to the imperfect correction of the cloud movement
during the TROPOMI-VIIRS measurement time difference, and to thin and/or small clouds resulting in false negative CF.
Again, the best performances are found for Southern Chile and Argentina, 3 August 2019, and for the Taklamakan desert, 22
December 2019, with an $\epsilon_O$ of 0.05 and 0.02, respectively.

## 420 5 Discussion

Here, we discuss some limitations and points of attention for the usage of the DARCLOS cloud shadow flags. Also, we point
out the possible spectral dependence of cloud shadow extents, and present the (unvalidated) spectral cloud shadow flag as an
auxiliary product of DARCLOS.

### 5.1 Limitations of the ACSF and PCSF

The PCSF depends on the CF which is determined by the effective cloud fraction. As discussed in Section 3.1, false negative
cloud detections in the CF can result in falsely detected gaps in cloud decks, resulting in false positive PCSFs inside the gaps
(Fig. 7a). Note that false negative cloud detections in the CF can also result in false *negative* shadow detections in the PCSF
and ACSF, since there is no shadow to be detected in the absence of a cloud detection. The surface albedo input for the effective
cloud fraction calculation in the $NO_2$ product is the LER climatology made by the Ozone Monitoring Instrument (OMI) at 440
nm available at a $0.5° \times 0.5°$ latitude-longitude grid (Kleipool et al., 2008). With a future implementation of the effective cloud





fraction from FRESCO which uses the TROPOMI DLER climatology at a $0.125° \times 0.125°$ latitude-longitude grid instead (see Sect. 2.2.2), the accuracy of the CF, PCSF and ACSF is expected to further increase.

DARCLOS has not been tested at regions covered by ice and/or snow, nor at sunglint geometries over ocean. In these circumstances, the performance of the current effective cloud fraction is limited, often resulting in false positive CFs. For the
ACSF, we have discarded the cloud pixels (and corresponding shadows) that contain a raised sunglint flag and/or snow/ice flag. For the PCSF, these pixels are not discarded, such that they are removed from the data when the PCSF and CF are used together to both remove cloud and cloud shadow contaminations.

The performance of the ACSF depends on the quality of the DLER climatology. Although the DLER takes into account monthly surface reflectivity changes throughout the year, temporary deviations from this climatology (e.g., agricultural land
usage changes, forest fires, precipitation, flooding and snow cover) are measured by the SCNLER, and may affect $\Gamma$ and possibly the ACSF. In addition, the spatial resolution of the DLER of $0.125° \times 0.125°$ is somewhat coarser than the spatial resolution of the TROPOMI Level 2 UVIS grid at which the SCNLER is measured. Dark small-scale surface features not captured by the DLER may, theoretically, give too low $\Gamma$ and may result in false positive ACSF. In the examples treated in this paper, however, dark small-scale forest (Sect. 3.2) and desert (Sect. 3.3) features did not convincingly deteriorate the ACSF
performance.

Both the irradiance and radiance measurements by TROPOMI have degraded during its operational lifetime. The irradiance measurements are known to degrade faster than the radiance measurements (and most significantly at the shortest wavelengths), leading to an increasing derived reflectance over time (Tilstra et al., 2020; Ludewig et al., 2020). Since the release of the version 2.0.0 TROPOMI level 1b processor on 5 July 2021, the irradiance degradation is being corrected. The thresholds used in this
paper for clouds and cloud shadows, which were set at an effective cloud fraction of 0.05 and $\Gamma = -15\%$ respectively, may have to be adjusted for the corrected data.

## 5.2 Spectral cloud shadow flag (SCSF)

In Sect. 4, the ACSF has been validated by visual inspection of true color images of the VIIRS instrument. Hence, the shadows found with the ACSF can be interpreted as the shadows that could be observed from space by the human eye. We find, however,
a significant wavelength dependency of the contrast parameter $\Gamma$ in the UV part of the spectrum. For example, Fig. 12a shows $\Gamma$ at 340 nm for the example above the Netherlands and Germany on 18 November 2018. Comparing to $\Gamma$ at $\lambda_{max}$ (Fig. 8b), where $\lambda_{max} = 772$ nm over land, the negative $\Gamma$ related to cloud shadows between 52°-53°N latitude and 9°-12°E longitude have disappeared. Also, the locations of some pixels with significant negative $\Gamma$ have changed. Although these shadows have not been validated (they could possibly not be observed by the human eye) or could be a result of noisy $\Gamma$ in dark scenes
(cf. Eq. (11)), they may be relevant for studying shadow effects on TROPOMI air quality products retrieved at particular UV wavelengths, such as the Absorbing Aerosol Index (AAI) at $\lambda = 340$ nm and $\lambda = 380$ nm (see de Graaf et al., 2005; Stein Zweers et al., 2018; Kooreman et al., 2020) or the $NO_2$ column at $\lambda = 440$ nm (see van Geffen et al., 2021). Therefore,



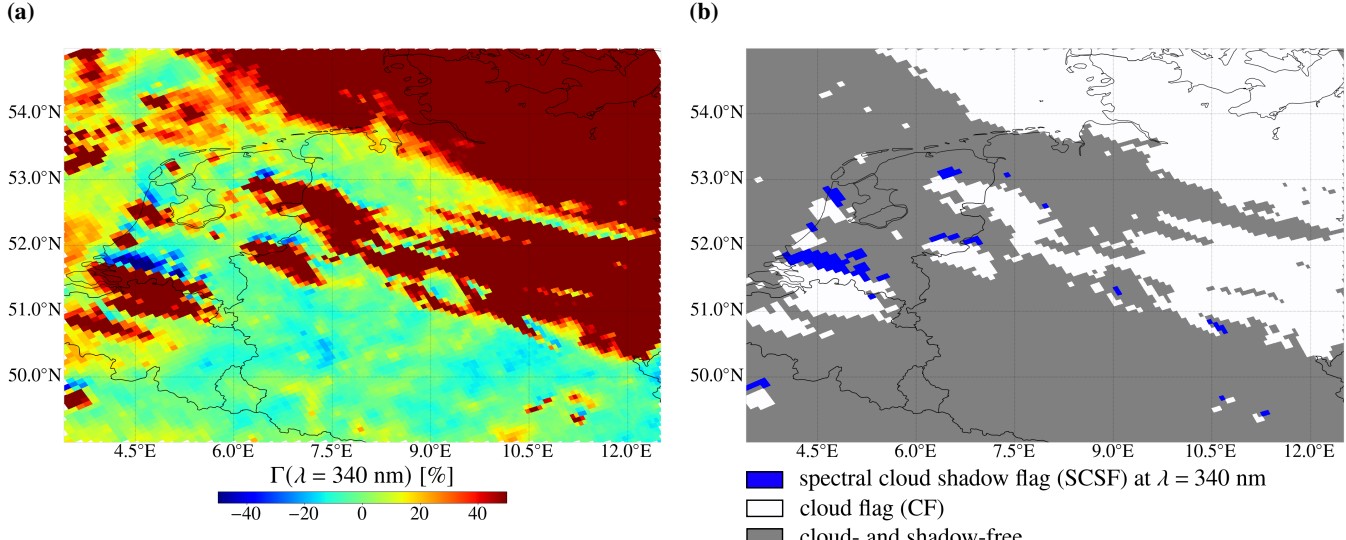

**Figure 12.** SCNLER-DLER contrast parameter $\Gamma$ at $\lambda = 340$ nm measured by TROPOMI (Fig. 12a) and TROPOMI Level 2 UVIS ground pixels with raised SCSFs at $\lambda = 340$ nm (Fig. 12b), for the Netherlands and Germany on 18 November 2018. In Fig. 12b, white pixels are cloud pixels and grey pixels do not contain a raised cloud or shadow flag.

DARCLOS also outputs the spectral cloud shadow flag (SCSF), which is raised at PCSF pixels for which:

$$\Gamma(\lambda) < q, \tag{17}$$

where $q$ is again set at $-15\%$. Contrary to the ACSF (Eq. 13), the SCSF is by definition wavelength dependent. The SCSF is computed at 328, 335, 340, 354, 367, 380, 388, 402, 416, 425, 440, 463, 494 nm.

Figure 12b shows the SCSF at $\lambda = 340$ nm for the example above the Netherlands and Germany on 18 November 2018. Comparing to Fig. 8d shows that part of the shadow flags has disappeared or has changed location. For example, the cloud shadow detected with the SCSF at $\sim 49.5°$N latitude and $\sim 11.3°$E longitude has shifted closer to the cloud as compared to the

corresponding ACSF shadow. We speculate that the wavelength dependence of shadow locations in the UV can be explained by the wavelength dependence of the molecular scattering optical thickness of the atmosphere: at shorter wavelengths, the molecular scattering optical thickness is higher such that higher atmospheric layers are probed from space, decreasing the observed shadow extents with TROPOMI. The explanation and validation of the wavelength dependence of observed cloud shadow extents in the UV is subject to future research.

**6 Summary and conclusions**

In this paper, we have demonstrated DARCLOS, a cloud shadow detection algorithm for TROPOMI. DARCLOS provides a potential cloud shadow flag (PCSF) based on geometric variables stored in TROPOMI Level 2 data, and an actual cloud shadow flag (ACSF) based on the contrast of the measured scene reflectivity with the climatological surface reflectivity. For





each TROPOMI pixel, this contrast is computed at the wavelength where the DLER is largest. The ACSFs are a subset of the
PCSFs.

Three case studies with different spectral surface albedo and cloud types have been discussed in detail. We have shown
that the PCSF vastly overestimates the shadows observed in true color images of the VIIRS-NPP instrument, as expected.
The shadows detected with the ACSF are better approximations of these true shadows, but may miss some shadows that are
produced by thin and/or small clouds. We showed that the shadow signatures in the contrast between the scene reflectivity
and the climatological surface reflectivity can, for almost all pixels, only be attributed to cloud shadows. That is, dark surface
features are not falsely detected as cloud shadows in the ACSF.

The PCSF and ACSF are validated by visual inspection of true color images made by the VIIRS-NPP instrument, for
in total six cases. We found that the cloud motion during the measurement time difference between TROPOMI and VIIRS
complicates this validation strategy. We showed that a cloud movement correction using the wind speed vectors at the cloud
height significantly improves the validation results. The best detection scores were achieved for the cases with relatively thick
and horizontally large cloud decks (ACSF $F_1$ score $\geq 0.94$ and PCSF omission error $\leq 0.05$). After the cloud movement
correction, the validation may still suffer from cloud evolution and the relatively coarse resolution of the wind product. Hence,
the true shadow detection performance at the TROPOMI measurement times may be expected to be even better than presented
with the validation in this paper.

DARCLOS is, to the best of our knowledge, the first cloud shadow detection algorithm for a spaceborne spectrometer in-
strument. In principle, DARCLOS can also be used for other spectrometer instruments than TROPOMI which have a spatial
resolution high enough to observe cloud shadows. An effective cloud fraction and climatological surface albedo are prerequi-
sites for DARCLOS, and should therefore be available at the ground pixel grid of the instrument. It should be noted that, when
computing the ACSF using UVIS and NIR wavelengths from different detectors, a co-registration of the SCNLER measure-
ments from one detector ground pixel grid to the other has to be performed to compute the ACSF. Ideally, true color images
are available of the scenes with approximately the same measurement times, in order to validate the detected cloud shadows
by visual inspection and to optimize the cloud and cloud shadow thresholds.

We conclude that the PCSF can be used to remove cloud shadow contaminated pixels from TROPOMI Level 2 UVIS data,
and that the actual cloud shadow flag can be used to select pixels for further analysis of cloud shadow effects. If both cloud and
cloud shadow effects are to be removed, the PCSF and CF can be used together. Also, the ACSF can be used to demonstrate
and/or count the true shadows that would be observed from space by the human eye. However, at UV wavelengths, we have
found indications of the wavelength dependence of cloud shadow signatures, and a spectrally dependent cloud shadow flag such
as the SCSF could possibly be more suitable when selecting shadow pixels in air quality products retrieved at UV wavelengths.
Further research is needed to explain and validate the spectral dependence of these cloud shadow signatures. The detection
of shadows with the ACSF and SCSF allows users to perform this analysis, and is a first step towards the understanding and
correction of cloud shadow effects on satellite spectrometer air quality measurements.



*Author contributions.* V.T. did all computations and wrote the manuscript. P.W. weekly commented on the intermediate results and guided V.T. to focus on the most relevant aspects. L.G.T. provided the SCNLER algorithm and commented on intermediate results. All authors read the manuscript, provided feedback that led to improvements and were involved in the selection of the results presented in this paper.

*Competing interests.* The authors declare that they have no conflict of interest.

*Acknowledgements.* This work is part of the research programme User Support Programme Space Research (GO) with project number ALWGO.2018.016, which is (partly) financed by the Dutch Research Council (NWO).



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
