# Peer review of "DARCLOS: a cloud shadow detection algorithm for TROPOMI"

_Atmospheric Measurement Techniques, 2021_

## Referee Comment (RC1)

Review of AMT_2021_377 "DARCLOS: a cloud shadow detection algorithm for TROPOMI" by Trees et al.

This paper discusses the DARCLOS cloud shadow detection algorithm, and applies it to TROPOMI radiances. The algorithm is clearly explained and the paper is well written, and should be published after minor revisions.

General comments

The DARCLOS algorithm relies on a longitude-latitude monthly climatology of cloud heights. The authors need to discuss the errors associated with the climatology that is applied in the paper. What are the standard deviations (state 1 or 2 sigma) of the cloud heights compared to cloud height validation data? The authors should discuss this by reference to the content in the Koelemeijer et al., 2001 and Wang et al., 2008 papers.

It is confusing to read on page 6, line 145 that "hc is the 145 FRESCO cloud height:, while on page 20, line 430 that "With a future implementation of the effective cloud fraction from FRESCO which uses the TROPOMI DLER climatology..". On page 6, line 145, add a phrase "applied using the DLER climatology (discussed below)" to tell the reader FRESCO currently uses DLER, and that the cloud fraction portion of FRESCO uses LER climatology (page 20, line 429, "The surface albedo input for the effective cloud fraction calculation in the NO2 product is the LER climatology").

On page 21, line 433, it is stated that "DARCLOS has not been tested at regions covered by ice and/or snow, nor at sunglint geometries over ocean." Over the ocean of course a longitude-latitude climatology of clear ocean is problematic since glint reflectance is dependent on the 10m ocean windspeed. For a given ocean scene, however, it is possible to create a PDF of radiances, from which a cloud radiance threshold can be calculated which can be used to identify clouds. Did you try such a technique in the development of DARCLOS? It would be useful in the Conclusions section to briefly discuss how you will treat ocean glint scenes in future developments.

Mention in the Conclusions if / how ACSF and SCSF data will be stored in TROPOMI data files. Will this be done in already existing files on in new separate data files?

Specific comments

The term "in close constellation with TROPOMI" could be reworded to "in close proximity to TROPOMI".

The term "raise" is a bit confusing since equation (1) "raises, alters the height of" h in proportion to hc-hsfc, while the algorithm "raises, identifies" PCSF to ACSF values. "Raise" is used with different meanings in the text. To lessen the confusion, it is suggested to revise the following phrases:
Page 1, line 6, revise to "DARCLOS raises (identifies) potential cloud shadow flags"
Page 5, line 128, revise to "with a raised (identified) cloud flag (CF) and.."
Page 6, line 147 to "which assigns the cloud height h proportional to hc -hsfc."
Page 7, line 171 to "in which PCSFs are to be raised (identified), based on"

Equation (1) has a C factor, set to 0.5. How was the value of 0.5 determined? How did the F1 scores vary as C varied? I did not see a discussion of C in Section 4, while line 148 on page 6 implies that this topic would be discussed in Section 4.

Page 2, line 35. How did the ground pixels change from 7.2 x 3.6 to 5.6 x 3.6 on 6 August 2019? The sentence implies that the actual physical dimensions changed. Please clarify.

Page 7, line 107. The phrase "inside but near the edges of the cloud pixel" was not clear in my first reading. The word "inside" makes sense if the cloud pixel is larger than the TROPOMI pixel size. An additional sentence is suggested to clarify the situation.

Page 7, line 172. The term "cloud-free" was at first confusing with regard to point Q in Figure 3, since point Q is shaded, but point Q is not untouched by cloud effects (it is in fact the cloud shadow). There are some readers who consider a "cloud-free" pixel to be a pixel not perturbed in radiance value by the presence of a cloud – which can yield a radiance enhancement (point O) or a radiance dimming (the point Q cloud shadow). An additional sentence can be added to clarify and lessen any confusion.

Page 9, line 231. Explain the rationale for using the "the 10% lowest SCNLER measurements".

Page 9,. Line 241. Specify what Adler is.

Page 10, line 250 – Page 11, line 283. Consider moving these lines to Page 12, line 290. I found this text to be out of place, and perhaps better placed in an organizational sense in the next Section.

Page 11, Figure 5. It would be helpful for the reader to have λmax identified in the figure caption for both panels.

Page 13, line 302. Revise to "Only a few shadows of small isolated clouds are detected by the ACSF".

Page 13, line 312. Replaced "temporarily" by "mischaracterized". You don't know if the error is due to a temporal problem, so "mischaracterized" is suggested.

Criteria

1. Does the paper address relevant scientific questions within the scope of AMT? Yes
2. Does the paper present novel concepts, ideas, tools, or data? The paper focuses on a technique which has not been previously discussed in the literature.
3. Are substantial conclusions reached? Yes
4. Are the scientific methods and assumptions valid and clearly outlined?  Yes
5. Are the results sufficient to support the interpretations and conclusions? Yes, the F1 test values in Table 1 are convincing.

6. Is the description of experiments and calculations sufficiently complete and precise to allow their reproduction by fellow scientists (traceability of results)? Yes

7. Do the authors give proper credit to related work and clearly indicate their own new/original contribution? Yes, the Introduction does a good review of the literature.

8. Does the title clearly reflect the contents of the paper? Yes

9. Does the abstract provide a concise and complete summary? Yes

10. Is the overall presentation well structured and clear? Overall, Yes. I do have one organizational suggestion (move Page 10, line 250 – Page 11, line 283 to Page 12, line 290).

11. Is the language fluent and precise? Overall, Yes. The comments above point out a few word choices (such as "raise") which can be altered and/or clarified.

12. Are mathematical formulae, symbols, abbreviations, and units correctly defined and used? Yes.

13. Should any parts of the paper (text, formulae, figures, tables) be clarified, reduced, combined, or eliminated? The General Comments section above points out a few places of suggested clarifications.

14. Are the number and quality of references appropriate? Yes

15. Is the amount and quality of supplementary material appropriate? Not applicable.

---

## Author Response (AR1)

**Response to comment of Anonymous Referee #1 on "DARCLOS: a cloud shadow detection algorithm for TROPOMI" by Victor Trees et al.**

Victor J. H. Trees [1,2], Ping Wang [1], Piet Stammes [1], Lieuwe G. Tilstra [1], David P. Donovan [1,2], and A. Pier Siebesma [1,2]

[1]Royal Netherlands Meteorological Institute (KNMI), De Bilt, the Netherlands
[2]Delft University of Technology, Delft, the Netherlands

**Correspondence:** Victor Trees (victor.trees@knmi.nl)

We thank the reviewer for his/her careful reading and for the comments and suggestions, which have improved the manuscript. Below, we give in *blue italic* the reviewer's comment, in black our response, in *black italic* copied text from the manuscript and in *red italic* the changed or new text in the manuscript.

*This paper discusses the DARCLOS cloud shadow detection algorithm, and applies it to TROPOMI radiances. The algorithm is clearly explained and the paper is well written, and should be published after minor revisions.*

*General comments*

*The DARCLOS algorithm relies on a longitude-latitude monthly climatology of cloud heights. The authors need to discuss the errors associated with the climatology that is applied in the paper. What are the standard deviations (state 1 or 2 sigma) of the cloud heights compared to cloud height validation data? The authors should discuss this by reference to the content in the Koelemeijer et al., 2001 and Wang et al., 2008 papers.*

DARCLOS does not use a climatology of cloud heights, but the TROPOMI L2 FRESCO cloud height (see line 145). We have
increased the cloud top height to calculate the PCSF, resulting in an overestimation of the shadowed area in the PCSF. In the ACSF and SCSF, the overestimation is removed using the SCNLER-DLER contrast.

*It is confusing to read on page 6, line 145 that "hc is the 145 FRESCO cloud height:, while on page 20, line 430 that "With a future implementation of the effective cloud fraction from FRESCO which uses the TROPOMI DLER climatology..". On page*
*6, line 145, add a phrase "applied using the DLER climatology (discussed below)" to tell the reader FRESCO currently uses DLER, and that the cloud fraction portion of FRESCO uses LER climatology (page 20, line 429, "The surface albedo input for the effective cloud fraction calculation in the NO2 product is the LER climatology").*

We agree with the reviewer that this sentence could cause confusion. The current version is not using DLER climatology, but LER climatology. A future version will use the DLER climatology. We have changed the sentence as follows:

line 430: *With a future implementation of the effective cloud fraction from FRESCO which uses the TROPOMI DLER climatology at a $0.125° × 0.125°$ latitude-longitude grid instead (see Sect. 2.2.2), the accuracy of the CF, PCSF and ACSF is expected to further increase.* –> "*With a future implementation of the TROPOMI DLER climatology, which uses a $0.125° × 0.125°$ latitude-longitude grid instead (see Sect. 2.2.2), in the effective cloud fraction algorithm, the accuracy of the CF, PCSF and ACSF is expected to further increase.*"

    *On page 21, line 433, it is stated that "DARCLOS has not been tested at regions covered by ice and/or snow, nor at sunglint geometries over ocean." Over the ocean of course a longitude-latitude climatology of clear ocean is problematic since glint reflectance is dependent on the 10m ocean windspeed. For a given ocean scene, however, it is possible to create a PDF of radiances, from which a cloud radiance threshold can be calculated which can be used to identify clouds. Did you try such a*

*technique in the development of DARCLOS? It would be useful in the Conclusions section to briefly discuss how you will treat ocean glint scenes in future developments.*

    We did not try adjusting the surface albedo climatology for FRESCO. Indeed, with an ocean surface reflectance calculation, the surface albedo could potentially be adjusted. However, the glint and a cloud can possibly be equally bright at some locations. We speculate that after a glint correction an overcorrection could take place such that some clouds would be interpreted as cloud-free. This could potentially be solved using a multi-wavelength approach, however, because this can be considered a problem to solve in the FRESCO algorithm instead of in the DARCLOS algorithm, we do not further elaborate on this in the paper. We added the following sentence to this paragraph:

    line 437: "*With future potential improvements of FRESCO above glint and snow/ice regions, DARCLOS could be tested above glint and snow/ice regions. Then, the DLER for snow/ice conditions (see Tilstra, 2022) should be employed in DARCLOS, and*

*possibly an ocean surface reflectance calculation can help distinguishing between clouds and the glint.*"

    *Mention in the Conclusions if / how ACSF and SCSF data will be stored in TROPOMI data files. Will this be done in already existing files on in new separate data files?*

    We added the following sentence to the conclusion:

line 495: "*The shadow flags of DARCLOS are planned for implementation in the TROPOMI L2 SCNLER product.*"

    *Specific comments*

    *The term "in close constellation with TROPOMI" could be reworded to "in close proximity to TROPOMI".*

We changed "in close constellation with" to "in close proximity to" everywhere in the text as suggested.

    *The term "raise" is a bit confusing since equation (1) "raises, alters the height of" h in proportion to hc-hsfc, while the algorithm "raises, identifies" PCSF to ACSF values. "Raise" is used with different meanings in the text. To lessen the confusion, it is suggested to revise the following phrases:*

We do not agree that the usage of the verb 'raise' is confusing, because from the subject of the sentence (the flags) we think that the meaning of the verb 'raise' is clear. Instead of replacing the verb 'raise' in the context of raising flags, we replaced the verb 'raise' for 'increase' in the context of increasing the cloud height. We adjusted the text as follows:

line 147: "*we have introduced the safety margin $C$ which raises the cloud*" –> "*we have introduced the safety margin $C$ which increases the cloud height*"

*Page 1, line 6, revise to "DARCLOS raises (identifies) potential cloud shadow flags"*

See previous comment.

*Page 5, line 128, revise to "with a raised (identified) cloud flag (CF) and.."*

See previous comment.

*Page 6, line 147 to "which assigns the cloud height h proportional to hc -hsfc."*

See previous comment.

*Page 7, line 171 to "in which PCSFs are to be raised (identified), based on"*

See previous comment.

*Equation (1) has a C factor, set to 0.5. How was the value of 0.5 determined? How did the F1 scores vary as C varied? I did not see a discussion of C in Section 4, while line 148 on page 6 implies that this topic would be discussed in Section 4.*

Line 148 should not imply that this topic would be discussed in Section 4, because the reference to Section 4 (placed after 'false negative shadow detections') was meant to direct the reader to the explanation of the definition of a false negative shadow detection, rather than to an analysis of the convergence of PCSF omission error versus safety factor $C$. We do not add the lower values of $C$ yielding higher PCSF omission errors to this paper, because that could confuse the reader. As explained on line 149, with $C = 0.5$ the number of underestimated maximum potential shadow extents (the omission error of the PCSF)

converged to a minimum. We changed the following sentence:

line 147: "*We set $C = 0.5$, for which the number of false negative shadow detections (see Sect. 4) resulting from underestimated maximum potential shadow extents converged to a minimum.*" –> "*We set $C = 0.5$, for which the number of false negative shadow detections (i.e. the omission error of the PCSF, see Sect. 4) resulting from underestimated maximum potential shadow extents converged to a minimum.*"

*Page 2, line 35. How did the ground pixels change from 7.2 x 3.6 to 5.6 x 3.6 on 6 August 2019? The sentence implies that the actual physical dimensions changed. Please clarify.*

We added a footnote to the sentence on line 35 with the following text:

"*The radiance co-addition time reduced from 1080 to 840 ms starting in orbit 9388. This resulted in a decrease of the minimal*

*along-track sampling distance from 7 km at nadir to 5.5 km at nadir (see Sect. 14 of Ludewig et al., 2020).*"

*Page 7, line 107. The phrase "inside but near the edges of the cloud pixel" was not clear in my first reading. The word "inside" makes sense if the cloud pixel is larger than the TROPOMI pixel size. An additional sentence is suggested to clarify the situation.*

We rephrased this sentence as follows:

line 166: "*Moreover, the unknown true horizontal and vertical cloud extents are projected inside but near the edges of the cloud pixel.*" –> "*Moreover, the actual projections of the unknown true horizontal and vertical cloud extents are located inside but near the edges of the cloud pixel.*"

*Page 7, line 172. The term "cloud-free" was at first confusing with regard to point Q in Figure 3, since point Q is shaded, but point Q is not untouched by cloud effects (it is in fact the cloud shadow). There are some readers who consider a "cloud-free" pixel to be a pixel not perturbed in radiance value by the presence of a cloud – which can yield a radiance enhancement (point O) or a radiance dimming (the point Q cloud shadow). An additional sentence can be added to clarify and lessen any confusion.*

We added the following words:

line 172: "*... we flag all the cloud-free ground pixels within or intersected by the triangle $OPQ$.*" –> "*... we flag all the cloud-free ground pixels (i.e. for which no CF is raised) within or intersected by the triangle $OPQ$.*"

*Page 9, line 231. Explain the rationale for using the "the 10% lowest SCNLER measurements".*

We changed the following sentence as follows:

line 231: "*In the DLER algorithm, the 10% lowest SCNLER measurements in the seasonal grid cell were used, and measurements containing aerosols or clouds were excluded (see Tilstra, 2021).*" –> "*In the DLER algorithm, an initial cloud screening was performed on the basis of NPP-VIIRS cloud information. After that, the 10% lowest SCNLER measurements in the seasonal grid cell were used which serves as a second-stage cloud filter, and measurements containing aerosols were excluded* 120 *(see Tilstra, 2022).*"

*Page 9,. Line 241. Specify what Adler is.*

We changed the description as follows:

line 241: "*The division by $A_{DLER}$*" –> "*The division by $A_{DLER}$ (the value of the DLER)* "

*Page 10, line 250 – Page 11, line 283. Consider moving these lines to Page 12, line 290. I found this text to be out of place, and perhaps better placed in an organizational sense in the next Section.*

We have introduced a new subsection at this location in the paper:

***2.2.4. Rationale behind the SCNLER-DLER contrast parameter***

We have changed the first sentence of this subsection as follows:

line 250: "*Here, we demonstrate the behavior of the variables used in Eqs. (11) to (13) with an example measurement.*" –> "*Here, we demonstrate the behavior of the variables in Eqs. (11) to (13) which determine the SCNLER-DLER contrast parameter $\Gamma$ with an example measurement.*"

*Page 11, Figure 5. It would be helpful for the reader to have lmax identified in the figure caption for both panels.*

$\lambda_{\mathrm{max}}$ is not indicated in Figure 5, because $\lambda_{\mathrm{max}}$ may vary per pixel. Indeed, this happens not to be the case for the pixels in Figure 5, as can be seen in Figure 6a (all land pixels have $\lambda_{\mathrm{max}} = 772$ nm, and all ocean pixels have $\lambda_{\mathrm{max}} = 402$ nm), however, indicating $\lambda_{\mathrm{max}}$ would suggest that $\lambda_{\mathrm{max}}$ does not vary per pixel, which may cause confusion. Therefore, we decided to keep Figure 5 as is.

*Page 13, line 302. Revise to "Only a few shadows of small isolated clouds are detected by the ACSF".*

We changed the text as suggested:

line 310: "*Only few shadows of small isolated clouds are detected by with the ACSF.*" –> "*Only a few shadows of small isolated clouds are detected by the ACSF.*"

*Page 13, line 312. Replace "temporarily" by "mischaracterized". You don't know if the error is due to a temporal problem, so "mischaracterized" is suggested.*

We adjusted the text as follows:

line 312: "*by a temporarily bright surface*" –> "*by bright surfaces*"

**References**

Ludewig, A., Kleipool, Q., Bartstra, R., Landzaat, R., Leloux, J., Loots, E., Meijering, P., van der Plas, E., Rozemeijer, N., Vonk, F., and Veefkind, P.: In-flight calibration results of the TROPOMI payload on board the Sentinel-5 Precursor satellite, Atmospheric Measurement Techniques, 13, 3561–3580, https://doi.org/https://doi.org/10.5194/amt-13-3561-2020, 2020.

Tilstra, L. G.: TROPOMI ATBD of the directionally dependent surface Lambertian-equivalent reflectivity, KNMI Report S5P-KNMI-L3-0301-RP, Issue 1.2.0, https://www.temis.nl/surface/albedo/tropomi_ler.php, [Online; accessed 7-February-2022], 2022.

**Response to comment of Anonymous Referee #2 on "DARCLOS: a cloud shadow detection algorithm for TROPOMI" by Victor Trees et al.**

Victor J. H. Trees [1,2], Ping Wang [1], Piet Stammes [1], Lieuwe G. Tilstra [1], David P. Donovan [1,2], and A. Pier Siebesma [1,2]

[1]Royal Netherlands Meteorological Institute (KNMI), De Bilt, the Netherlands
[2]Delft University of Technology, Delft, the Netherlands

**Correspondence:** Victor Trees (victor.trees@knmi.nl)

We thank the reviewer for his/her careful reading and for the comments and suggestions, which have improved the manuscript. Below, we give in *blue italic* the reviewer's comment, in black our response, in *black italic* copied text from the manuscript and in *red italic* the changed or new text in the manuscript.

*General Comments*

*The manuscript presents a scheme for the detection of cloud shadows in observations made by the spaceborne imaging spectrometer TROPOMI. The scheme is new in the sense that it is based on the measurements of a spectrometer (rather than on multi-spectral measurements from imagers). Undetected cloud shadows can cause significant biases in the TROPOMI L2*
*products. The flag produced by the scheme enables the analysis of such biases and the masking of affected observations and is therefore of interest to the remote sensing community. The description of the scheme is concise and clear up to a few items listed below. The testing and validation of the scheme with imager data is adequate to showcase the performance of the scheme and is well presented.*

*Specific Comments*

1. *Novelty*

   *It is reported that heritage cloud shadow detection algorithms often use a combination of geometric and spectral schemes. The new scheme described in the present manuscript follows this strategy and is not new in that sense. Please clarify, probably best in the introduction, in which sense(s) the new scheme is different from heritage schemes.*

DARCLOS exploits the spectral resolution of TROPOMI for computing the ACSF, as it uses the wavelength for shadow detection where the surface reflectance is strongest. For high spectrally varying reflectors, this means that DARCLOS can choose the wavelength in the spectrum (out of a large sets of wavelengths) where the most stable shadow detection is expected.

The spectral cloud shadow flag (SCSF) detects another type of shadow (see Sec. 5.2): the wavelength dependent shadow. The length of this shadow is not necessarily the same as the shadow observed by an imager. We found a wavelength dependence of shadow signature locations in the UV (see Fig. 12). We speculate that, because the gas scattering optical thickness decreases with $\lambda^{-4}$, at shorter wavelengths higher layers of the atmosphere are probed in which shadows may be geometrically shorter. With the SCSF, we obtain a shadow flag dedicated to specific UV wavelengths where air quality products are retrieved (e.g., 340 and 380 nm for the AAI, and 440 nm for $NO_2$). Such a cloud shadow detection at the precise wavelengths of the spectrometer's air quality products is unique for DARCLOS and cannot be done with data from an imager. We changed the following:

line 100: "*As TROPOMI is a spectrometer, DARCLOS exploits the spectral ranges of TROPOMI by searching in each pixel for the most optimal wavelength for shadow detection independent of surface classification. The spectral tests are only based on the darkness of shadows relative to the reference data. This means that no assumptions are made about the color of cloud shadows.*" –> "*The spectral tests are only based on the darkness of shadows relative to the reference data. This means that no assumptions are made about the color of cloud shadows. As TROPOMI is a spectrometer, DARCLOS exploits the spectra of TROPOMI by using the wavelength for shadow detection where the surface reflectance is strongest, independent of surface classification. We validate the PCSF and ACSF with true color images of Suomi NPP VIIRS which orbits in close constellation with TROPOMI. Because geometrical shadow extents may be wavelength dependent, DARCLOS also outputs a wavelength dependent cloud shadow flag for the wavelengths at which TROPOMI's air quality products are retrieved. Such a cloud shadow detection at the precise wavelengths of TROPOMI's air quality products is unique for DARCLOS and cannot be done with data from an imager.*"

We added the following paragraph to the conclusion:

"*At UV wavelengths, we have found cloud shadow signatures at different locations than determined with the ACSF, potentially indicating a wavelength dependence of cloud shadow extents. Because TROPOMI's air quality products are retrieved at specific wavelengths or wavelength ranges, DARCLOS also outputs the spectral cloud shadow flag (SCSF), which is a wavelength dependent alternative for the ACSF. Although the SCSF may not be retrieved at the wavelength where the most stable wavelength independent (visible) shadow detection is expected, it may be a better estimate of the cloud shadow locations at the specific UV wavelengths of interest. Such a cloud shadow detection at the precise wavelengths of TROPOMI's air quality products is unique for DARCLOS and cannot be done with data from an imager.*"

We adjusted the abstract as follows:

"*[...] DARCLOS raises potential cloud shadow flags (PCSFs), actual cloud shadow flags (ACSFs) and spectral cloud shadow flags (SCSFs). The PCSFs indicate the TROPOMI ground pixels that are potentially affected by cloud shadows based on a geometric consideration with safety margins. The ACSFs are a refinement of the PCSFs using spectral reflectance information of the PCSF pixels, and identify the TROPOMI ground pixels that are confidently affected by cloud shadows. Because we find indications of the wavelength dependence of cloud shadow extents in the UV, the SCSF is a wavelength dependent alternative for the ACSF at the wavelengths of TROPOMI's air quality retrievals. We*

*validate the PCSF and ACSF with true color images made by the VIIRS instrument on board of Suomi NPP orbiting in close constellation with TROPOMI on board of Sentinel 5-P. We find that the cloud evolution during the overpass time difference of TROPOMI and VIIRS complicates this validation strategy, implicating that an alternative cloud shadow detection approach using colocated VIIRS data would be inaccurate. We conclude that the PCSF can be used to exclude cloud shadow contamination from TROPOMI data, while the ACSF and SCSF can be used to select pixels for the scientific analysis of cloud shadow effects."*

To the summary diagram (Figure 1), we added "*Spectral cloud shadow flag (SCSF)*" to the last grey box.

We added to the introduction of the Method section:

line 114: "*The spectral cloud shadow flag (SCSF) is a wavelength dependent alternative for the ACSF and will be explained in Sect. 5.*"

2. *Strategy*

*While it is stated that the scheme is the first one that works on spectrometer measurements (rather than on multi-spectral measurements from imagers), it does not exploit the high spectral resolution capability of the spectrometer. For TROPOMI observations, co-located VIIRS imager data are available with observation time differences of a few minutes. Therefore is seems valid to consider an alternative approach applying a performant cloud shadow detection algorithm to VIIRS data. Please discuss the benefits (eg availability of additional TIR information, better spatial resolution, wrt TROPOMI) and drawbacks (eg changes in clouds within the observation time difference (now discussed in the context of validation), dependence on another sensor and processing chain) of this alternative approach.*

DARCLOS does exploit the spectral resolution capability of TROPOMI for computing the ACSF and SCSF, as explained in the answer to previous comment. The spatial size of cloud shadows in TROPOMI data is 1 or several TROPOMI pixels. In the Validation Section, it was explained that clouds can change shape, appear, disappear and can shift at least 1 TROPOMI pixel during the VIIRS-TROPOMI measurement time difference interval for high clouds (and cloud shadows are particularly detectable from space when clouds are high). That is, the possible spatial error due to the cloud evolution is of the same order of magnitude as the spatial accuracy needed for shadow detection. Therefore, we find it fundamentally not accurate to use VIIRS measurements for shadow detection in TROPOMI data. This point was raised in the footnote on page 19.

We added to the abstract:

"*We find that the cloud evolution during the overpass time difference between TROPOMI and VIIRS complicates this validation strategy, implicating that an alternative cloud shadow detection approach using co-located VIIRS observations could be problematic.*"

3. *Performance*

*The performance is reported in terms of omission and commission errors and a derived score without reference to the performance of other cloud shadow flags. Please discuss the performance also in the context of the comparable products, as far as such performance data is available.*

We added the following paragraph to the Validation section:

*In order to put the validation results in perspective, we note that the state-of-the art imager cloud and cloud shadow detection code Fmask version 4.0 (Qiu et al., 2019) reports shadow detection commission errors of 0.49 for Landsat 4-7 and 0.38 for Landsat 8, and omission errors of 0.27 for Landsat 4-7 and 0.31 for Landsat 8. Using multi-temporal reference images of specific regions, Candra et al. (2019) achieved omission and commission errors ranging from 0.001 to 0.084 and 0 to 0.058, respectively, depending on the region. The PCSF omission errors and ACSF commission errors in Table 1 are lower than those of Fmask 4.0, and are of the same order of magnitude as those achieved by Candra et al. (2019). Of course, because of the much higher spatial resolution of Landsat than that of TROPOMI, the error values for Landsat actually refer to a much larger number of pixels.*

*Technical corrections*

*The processing flow chart (Figure 1) is inaccurate. Please distinguish data and processing steps clearly; Identify input data and output data, per processing step; Identify which parameters is passed on from one processing step to the next; distinguish climatological input from dynamic input from TROPOMI observations.*

Figure 1 is supposed to give a summary of the input and output data of DARCLOS. For readers that do not go through the technical details of the paper, Figure 1 is still readable as is. We incorrectly named Figure 1 'flow diagram'. A more suitable name would be 'summary of inputs and outputs'. We changed the following:

caption of Figure 1: "*Flowchart of the algorithm.*" –> "*Summary of the inputs and outputs of DARCLOS.*"

line 109: "*The flowchart in Fig. 1 summarizes the algorithm setup and serves as a road map for this section.*" –> "*Figure 1 summarizes the inputs and outputs of DARCLOS.*"

*In the introduction (Section 2.2, line 194) it is stated that the actual cloud shadow flag (ACSF) is raised "based on the darkness of the shadowed pixels with respect to non-shadowed pixels", which suggests that multiple pixels in a field of regard are evaluated, for each pixel. In contrast, according to Section 2.2.3, the ACSF is raised based on radiometric criteria for each pixel independently. Please clarify and align.*

We thank the reviewer for pointing out this confusing formulation. The ACSF is indeed raised based on radiometric thresholds. We found it most clear for the reader to shorten this paragraph, since this level of detail is not necessary in this introductory part of the Section:

line 192: "*Then, we compare this corrected reflectance to the expected surface reflectance from climatological observations by TROPOMI, revealing the actual shadowed pixels. The ACSF determination is based on the darkness of the shadowed pixels with respect to non-shadowed pixels, which is most apparent at the wavelength where the surface reflectance is strongest.*" –> "*Then,*

*we compare the corrected reflectance to the expected surface reflectance from climatological observations by TROPOMI, revealing the actual shadowed pixels. This comparison is done at the wavelength where the surface reflectance is strongest.*"

**References**

Candra, D. S., Phinn, S., and Scarth, P.: Automated Cloud and Cloud-Shadow Masking for Landsat 8 Using Multitemporal Images in a Variety of Environments, Remote Sensing, 11, 2060, https://doi.org/10.3390/rs11172060, 2019.

Qiu, S., Zhu, Z., and He, B.: Fmask 4.0: Improved cloud and cloud shadow detection in Landsats 4-8 and Sentinel-2 imagery, Remote Sensing of Environment, 231, 111 205, https://doi.org/10.1016/j.rse.2019.05.024, 2019.

---

## Referee Report (RR1)

Review of AMT_2021_377 "DARCLOS: a cloud shadow detection algorithm for TROPOMI"
by Trees et al.

The revised paper discusses the DARCLOS cloud shadow detection algorithm, and applies it to TROPOMI radiances. The algorithm is clearly explained and the paper is well written, and should be published after minor suggested clarifications.

General points

Page 23, line 500
If the information is available, specify the general pathnames of the TROPOMI level 2 files for which the PCSF, ACSF, and SCSF flags are included, and specify the variable names that are used in the files.

Page 23, line 505
Specify in the text the percent frequency of occurrence for which the PCSF overestimates cloud shadows.

Minor points

In the Figure 5 caption the sentence "Here, all measurements are cloud-free" is ambiguous (not clear). Please clarify.

Page 12, line 298.
Suggest to replace with "introduced above (Sec. 3.1)"

The longitude-latitude entries in Table 1 have a different range than the Figures. Clarify in the text.

Criteria

1. Does the paper address relevant scientific questions within the scope of AMT? Yes

2. Does the paper present novel concepts, ideas, tools, or data? Yes, the authors point out that DARCLOS is the first cloud shadow detection algorithm for a spaceborne spectrometer.

3. Are substantial conclusions reached? Yes

4. Are the scientific methods and assumptions valid and clearly outlined? Yes

5. Are the results sufficient to support the interpretations and conclusions? Yes

6. Is the description of experiments and calculations sufficiently complete and precise to allow their reproduction by fellow scientists (traceability of results)? Yes

7. Do the authors give proper credit to related work and clearly indicate their own new/original contribution? Yes, the Introduction does a good review of the literature.

8. Does the title clearly reflect the contents of the paper? Yes

9. Does the abstract provide a concise and complete summary? Yes

10. Is the overall presentation well-structured and clear? Yes

11. Is the language fluent and precise? Yes.

12. Are mathematical formulae, symbols, abbreviations, and units correctly defined and used? Yes.

13. Should any parts of the paper (text, formulae, figures, tables) be clarified, reduced, combined, or eliminated? The comments above discuss a few minor suggested clarifications.

14. Are the number and quality of references appropriate? Yes

15. Is the amount and quality of supplementary material appropriate? Not applicable.

---

## Author Response (AR2)

**Response to comment of Anonymous Referee #3 on "DARCLOS: a cloud shadow detection algorithm for TROPOMI" by Victor Trees et al.**

Victor J. H. Trees [1,2], Ping Wang [1], Piet Stammes [1], Lieuwe G. Tilstra [1], David P. Donovan [1,2], and A. Pier Siebesma [1,2]

[1]Royal Netherlands Meteorological Institute (KNMI), De Bilt, the Netherlands
[2]Delft University of Technology, Delft, the Netherlands

**Correspondence:** Victor Trees (victor.trees@knmi.nl)

We thank the reviewer for his/her careful reading and for the comments and suggestions, which have improved the manuscript. Below, we give in *blue italic* the reviewer's comment, in black our response, in *black italic* copied text from the manuscript and in *red italic* the changed or new text in the manuscript.

5 *General points*

- *Page 23, line 500: If the information is available, specify the general path names of the TROPOMI level 2 files for which the PCSF, ACSF, and SCSF flags are included, and specify the variable names that are used in the files.*

  The path names and final variable names of the shadow flags in the TROPOMI level 2 files are not yet determined.

- *Page 23, line 505: Specify in the text the percent frequency of occurrence for which the PCSF overestimates cloud shadow*

10 We have added the PCSF commission error (representing the overestimation by the PCSF) to Table 1, and modified the following sentence in the text as follows:

  Page 20, line 426: *"The fourth and fifth column of Table 1 show $\epsilon_O$ and $\epsilon_C$, respectively, of the PCSF. The value of $\epsilon_C$ of the PCSF is higher than 0.48 and higher than that of the ACSF, ..."*

*Minor points*

15 - *In the Figure 5 caption the sentence "Here, all measurements are cloud-free" is ambiguous (not clear). Please clarify.*
  We modified this sentence as follows:
  *Here, all measurements are cloud-free.* –> *Here, all measurements are cloud-free (i.e., without CF).*

- *Page 12, line 298. Suggest to replace with "introduced above (Sec. 3.1)"*
  We modified this sentence as follows:
20 *"introduced above (Sec. 3.1)"* –> *"introduced in Sect. 2.2.4 (Sec. 3.1)"*

– *The longitude-latitude entries in Table 1 have a different range than the Figures. Clarify in the text.*

We have added the following to line 394:

*"For the ACSF in a slightly larger area than shown in Fig. 11a (36.5 - 43.0 °N, 76.0 - 88.0 °E), ... "*